# Analysis of Fourier Neural Operators via Effective Field Theory

## Abstract

Fourier Neural Operators (FNOs) have emerged as leading surrogates for solver operators for various functional problems, yet their stability, generalization and frequency behavior lack a principled explanation. We present a systematic effective field theory analysis of FNOs in an infinite dimensional function space, deriving closed recursion relations for the layer kernel and four-point vertex and then examining three practically important settings—analytic activations, scale invariant cases and architectures with residual connections. The theory shows that nonlinear activations inevitably couple frequency inputs to high frequency modes that are otherwise discarded by spectral truncation, and experiments confirm this frequency transfer. For wide networks, we derive explicit criticality conditions on the weight initialization ensemble that ensure small input perturbations maintain a uniform scale across depth, and we confirm experimentally that the theoretically predicted ratio of kernel perturbations matches the measurements. Taken together, our results quantify how nonlinearity enables neural operators to capture non-trivial features, supply criteria for hyperparameter selection via criticality analysis, and explain why scale invariant activations and residual connections enhance feature learning in FNOs.

## 1 Introduction

### 1.1 Fourier Neural Operator

In scientific machine learning, neural operators—networks that approximate solution operators so they can act directly on functional data—are attracting growing attention (Kovachki et al. (2021b), Lu et al. (2021)). In addition to the linear layers of standard fully connected networks (FCNs), neural operator architectures include layers that approximate a kernel and apply an integral transform. How the kernel is represented gives rise to many variants: graph based constructions (Li et al. (2020)), tensor product decompositions (Kovachki et al. (2021b)), hierarchical (multi-resolution) kernels (Gupta et al. (2021)), and, most notably, Fourier Neural Operators (FNOs) (Li et al. (2021)). Thanks to their computational efficiency, high accuracy, and ability to capture long range interactions that elude CNNs and GNNs, FNOs have become the workhorse choice for surrogate modeling (Pathak et al. (2022), Sun et al. (2023)). Prior theory has addressed their generalization error (Kim & Kang (2024), Benitez et al. (2024)), the universal approximation property (Kovachki et al. (2021a), Lee et al. (2025)), and expressivity/trainability from a mean-field perspective (Koshizuka et al. (2024)), yet a comprehensive statistical description is still lacking. Here we fill that gap by analyzing FNOs through the lens of effective field theory (EFT).

### 1.2 Effective Field Theory for Neural Networks

Because stochastic elements such as random initialization and stochastic gradient descent are intrinsic to neural networks, statistical-physics tools are natural candidates for theoretical analysis. Recent work has imported methods from field theory (Halverson et al. (2021), Banta et al. (2024)), relating connected correlators in randomly initialized ensembles and deriving susceptibility based criticality conditions that predict when training remains stable (Roberts et al. (2022)). Within the EFT framework one can see how the choice of activation or hyperparameters moves a model between different universality classes, revealing when signals explode, vanish, or propagate cleanly.

### 1.3 CONTRIBUTIONS

Building on the EFT formulation for FCNs, we extend the approach to Fourier Neural Operators—an essential step, because FNOs act on infinite dimensional function spaces rather than finite-dimensional vectors. While the basic relations among connected correlators mirror those in FCNs, the architectural differences introduce qualitatively new phenomena: the observables become functions rather than scalars, and mechanisms such as frequency coupling appear. We show that the kernel function—the infinite width approximation of the two-point correlator—admits a closed layerwise recursion, and we derive analytic formulas for kernel perturbations; we then validate these predictions experimentally. Our analysis quantifies how non-trivial activations and residual connections influence information flow in FNOs and yields explicit design criteria for stable models.

## 2 PRELIMINARIES

In this section, we examine the definition of neural operators designed for processing functional data, and explore effective field theory along with its application to the statistical analysis of basic neural networks (fully connected networks). Henceforth, the Einstein summation convention will be adopted for repeated indices.

### 2.1 EFFECTIVE FIELD THEORY

Suppose that the distribution $p(X_1, \ldots, X_n) \sim \exp(-S(X_1, \ldots, X_n))$ is given. For an analytic function $f(x_1, \ldots, x_n) = \sum a_{i_1 \ldots i_n} x_1^{i_1} \ldots x_n^{i_n}$ the expectation of $f$ with respect to $p$ is expressed as

$$\mathbb{E}[f(X_1, \ldots, X_n)] = \sum a_{i_1 \ldots i_n} \mathbb{E}[X_1^{i_1} \ldots X_n^{i_n}]$$

Thus, in the analytic observable case, the collection of terms $\mathbb{E}[X_1^{i_1} \ldots X_n^{i_n}]$ contains all the information about the distribution. These terms are referred to as the $(i_1 + \cdots + i_n)$-point correlators (or $(i_1 + \cdots + i_n)$-th moments). For a Gaussian distribution, where $S(X_1, \ldots, X_n)$ consists solely of quadratic terms, the following well-known result holds:

**Proposition 1 (Wick contraction)** Suppose $(X_1, \ldots, X_n)$ is a zero-mean multivariate normal random vector. Then, all odd order correlators vanish, and the even order correlators are given by

$$\mathbb{E}[X_{j_1}^{i_1} \ldots X_{j_n}^{i_n}] = \sum_{\text{(all possible pairings)}} \prod_{\text{(pairings)}} \mathbb{E}[X_{p_1} X_{p_2}]$$

Motivated by this proposition, we define the connected correlator (or cumulant) for an arbitrary distribution as follows:

**Definition 1** The $k$-point connected correlator (or $k$-th cumulant, $k = i_1 + \cdots + i_n$) is defined by

$$\mathbb{E}|_{\text{conn}}[X_{j_1}^{i_1} \ldots X_{j_n}^{i_n}] := \mathbb{E}[X_{j_1}^{i_1} \ldots X_{j_n}^{i_n}]$$
$$- \sum_{\text{all subdivisions of } (1, \ldots, k)} \mathbb{E}[X_{j_{\mu_1,1}} \ldots X_{j_{\mu_1,t_1}}]|_{\text{conn}} \ldots \mathbb{E}[X_{j_{\mu_\nu,1}} \ldots X_{j_{\mu_\nu,t_\nu}}]|_{\text{conn}}$$

Since the connected correlators vanish for Gaussian distributions, they serve as an indicator of the deviation of a distribution from Gaussianity. A review of the effective theory analysis for fully connected networks (FCNs) is provided in Appendix A.

### 2.2 NEURAL OPERATOR

The architecture of the neural operator we consider consists of the following two types of layers: Eq. (1) describes a linear layer that linearly transforms the channel dimension of functional data, and Eq. (2) represents a kernel integration layer performing convolution operations with a kernel function. Eq. (3) expresses the kernel integration layer combined with a non-linear activation function. In general, the kernel function does not need to be translationally invariant; however, in our setting, we focus on the translationally invariant case as expressed in Eq. (2).

$$\left( \mathcal{L}(u)(x) \right)_i := \sum_j W_{ij} u_j(x), \quad i = 1, \ldots, n. \tag{1}$$

$$\Big(\mathcal{R}(u)(x)\Big)_i := \int_{\mathbb{R}^d} k_{ij}(x - x')u_j(x')dx', \quad i = 1, \ldots, n. \tag{2}$$

$$\mathfrak{S}(x) := \sigma(\mathcal{R}(u)(x)). \tag{3}$$

**Definition 2** By composing the linear layers and kernel integration layers introduced above, we build the Neural Operator architecture as follows:

$$\begin{aligned} \mathcal{Z}^{(1)} &:= \mathcal{L}_{\text{lift}}(u), \\ \mathcal{Z}^{(l+1)}(u) &:= \mathcal{R}^{(l+1)}\Big(\mathfrak{S}^{(l)}(u)\Big), \quad l = 1, \ldots, L - 1, \\ \mathcal{Y}(u) &:= \mathcal{L}_{\text{proj}}\Big(\mathcal{Z}^{(L)}(u)\Big). \end{aligned} \tag{4}$$

Here, $\mathcal{L}_{\text{lift}}$ and $\mathcal{L}_{\text{proj}}$ are linear layers. $\mathcal{L}_{\text{lift}}$ lifts the input function's pointwise values into a higher dimensional feature space, $\mathcal{L}_{\text{proj}}$ projects the pre-activations—after they have passed through the sequence of kernel integration layers—back down to the dimensionality of the target function. $\mathcal{R}^{(k)}$ denotes a kernel integration layer. In the Fourier Neural Operator of Li et al. (2021), each kernel integration layer is implemented by taking the Fourier transform of its input, multiplying by a learnable tensor, and then applying the inverse Fourier transform. Concretely, one can write:

$$(u_i^{(k)}) \mapsto \sigma\Big(\mathcal{F}^{-1}\Big(\sum_j R_{ij}^{(k+1)}(f)\hat{u}_j^{(k)}(f)\Big)\Big)$$

where $R_{ij}^{(k+1)}(f)$ represents the parameterized complex-valued tensor. We adopt the convention of using $\mathcal{F}$ and the hat notation for the Fourier transform, and $\mathcal{F}^{-1}$ for the inverse transform. As with fully-connected networks, we initialize all parameters so that they follow the statistical distributions:

$$\begin{aligned} \mathbb{E}[R_{ij}^{(k+1)}(f)] &= 0, \\ \mathbb{E}[\text{Re}(R_{i_1 j_1}^{(k+1)}(f))\text{Re}(R_{i_2 j_2}^{(k+1)}(f'))] &= \frac{C_{R^{(k+1)}}(f)}{2n^{(k)}}\delta_{i_1 i_2}\delta_{j_1 j_2}\delta(f - f'), \\ \mathbb{E}[\text{Im}(R_{i_1 j_1}^{(k+1)}(f))\text{Im}(R_{i_2 j_2}^{(k+1)}(f'))] &= \frac{C_{R^{(k+1)}}(f)}{2n^{(k)}}\delta_{i_1 i_2}\delta_{j_1 j_2}\delta(f - f'), \\ \mathbb{E}[\text{Re}(R_{i_1 j_1}^{(k+1)}(f))\text{Im}(R_{i_2 j_2}^{(k+1)}(f'))] &= 0. \end{aligned} \tag{5}$$

Therefore, the initialization distribution of the kernel integration layers can be viewed as white noise, and when implemented in practice via discretization, it matches the setup in Li et al. (2021). Our neural-operator architecture assumes real-valued functions. To guarantee that multiplying by the parameter tensor $R_{ij}^{(k)}$ and then applying the inverse Fourier transform still yields a real function, we theoretically impose the symmetry on the sampled parameters: $R_{ij}^{(k)}(f) = \overline{R_{ij}^{(k)}}(-f)$. During training the values of $R_{ij}^{(k)}$ evolve, yet the symmetry is preserved in practice because the implementation processes data with a real discrete Fourier transform, which enforces the required conjugate symmetry automatically. For the linear layers, we define their initialization distributions in exactly the same way as FCN in Appendix A.

## 3 EFFECTIVE FIELD THEORY FOR NEURAL OPERATORS

Following the effective field theory (EFT) framework for neural networks set out in Appendix A, the functional degrees of freedom $u$ are assumed to follow the Boltzmann type distribution

$$P(u) \propto \exp(-S(u)).$$

where the action $S(u)$ fully determines the statistical weight of each configuration. The expectation value of $u$ at position $x$ is therefore obtained from the path integral

$$\mathbb{E}[u(x)] = \frac{\int u(x)P(u)\mathcal{D}u}{\int P(u)\mathcal{D}u}.$$

Because both differentiation and ordinary integration act linearly on the functional measure, connected correlators furnish a systematic means of computing expectation values that involve products, derivatives, or integrals of multiple fields. In what follows, closed form expressions for these correlators are derived, and—paralleling the analysis in Appendix A—recursive relations are established for the two-point kernel and the four-point vertex. Finally, explicit formulas are provided for the susceptibility of the kernel to perturbations parallel and perpendicular to a chosen reference trajectory in function space, thereby quantifying anisotropic responses to input variations.

## 3.1 CORRELATORS FOR NEURAL OPERATORS

Henceforth, boldface letters indicate vector-valued functions. And notation $\hat{u}$ also means $\mathcal{F}(u)$. Because the explicit evaluation of correlators in layers that include convolution is prohibitively cumbersome, we instead work in Fourier space and compute the correlators of the Fourier transformed pre-activations. In particular, Eq. (6) gives the two-point correlator of the $l$-th layer conditioned on the $(l-1)$-th layer's outputs $\mathbf{u}$ and $\mathbf{v}$. We write $\left( \mathcal{Z}^{(l)}|_{\mathbf{u}} \right)_i$ for the $i$-th component of $\mathcal{Z}^{(l)}|_{\mathbf{u}}$, that is, the $i$-th pre-activation in layer $l$ when the preceding layer is fixed to the function $\mathbf{u}$. Likewise, we write $\left( \mathcal{Z}^{(l)}\{\mathbf{u}\} \right)_i$ for the corresponding output component of layer $l$ when the input function is $\mathbf{u}$.

$$
\mathbb{E}\left[ \mathcal{F}\left( \mathcal{Z}^{(l)}|_{\mathbf{u}} \right)_i(f)\overline{\mathcal{F}\left( \mathcal{Z}^{(l)}|_{\mathbf{v}} \right)}_{i'}(f') \Big| u, v \right]
$$
$$
= \mathbb{E}\left[ \sum_{jj'} R_{ij}^{(l)}(f)\overline{R_{i'j'}^{(l)}(f')}\hat{u}_j(f)\overline{\hat{v}_{j'}}(f') \Big| u, v \right]
$$
$$
= \sum_{jj'} \frac{C_{R^{(l)}}}{n^{(l-1)}}\delta(f - f')\delta_{ii'}\delta_{jj'}\hat{u}_j(f)\overline{\hat{v}_{j'}}(f')
$$
$$
= \frac{C_{R^{(l)}}}{n^{(l-1)}}\delta(f - f')\delta_{ii'}\sum_j \hat{u}_j(f)\overline{\hat{v}_j}(f'). \tag{6}
$$

Although we currently compute all statistical objects for the Fourier transforms of the pre-activation functions, the corresponding quantities in the spatial domain are recovered via Eq. (7). Specifically,

$$
\mathbb{E}\left[ \left( \mathcal{Z}^{(l)}|_{\mathbf{u}}(x) \right)_i\overline{\left( \mathcal{Z}^{(l)}|_{\mathbf{v}}(y) \right)}_{i'} \Big| \mathbf{u}, \mathbf{v} \right]
$$
$$
= \int_{\mathbb{R}^d} \int_{\mathbb{R}^d} \mathbb{E}\left[ \mathcal{F}\left( \mathcal{Z}^{(l)}|_{\mathbf{u}} \right)_i(f)\overline{\mathcal{F}\left( \mathcal{Z}^{(l)}|_{\mathbf{v}} \right)}_{i'}(f') \Big| \mathbf{u}, \mathbf{v} \right] e^{ifx}e^{-if'y}df df' \tag{7}
$$
$$
= \frac{C_{R^{(l)}}}{n^{(l-1)}}\delta_{ii'}\mathcal{F}^{-1}\left( \langle \hat{\mathbf{u}}, \hat{\mathbf{v}} \rangle \right).
$$

Now, we consider two-point correlator for neural operator, since $l$-th layer parameters and variables up to $(l-1)$-th layers are independent, the two-point correlator of a neural operator factorizes as Eq. (8).

$$
\mathbb{E}\left[ \mathcal{F}\left( \mathcal{Z}^{(l)}\{\mathbf{u}\} \right)_i(f)\overline{\mathcal{F}\left( \mathcal{Z}^{(l)}\{\mathbf{v}\} \right)}_{i'}(f') \right]
$$
$$
= \frac{C_{R^{(l)}}}{n^{(l-1)}}\delta_{ii'}\delta(f - f')\mathbb{E}\left[ \left( \mathcal{F}\left( \mathfrak{S}^{(l-1)}\{\mathbf{u}\} \right)(f) \cdot \overline{\mathcal{F}\left( \mathfrak{S}^{(l-1)}\{\mathbf{v}\} \right)}(f') \right) \right]. \tag{8}
$$

and we define the $l$-th layer mean metric $\mathcal{G}^{(l)}$ as non-trivial part of Eq. (8) as Eq. (9) and stochastic metric $\widetilde{\mathcal{G}^{(l)}}$ as Eq. (10) in which expectation is not taken except on last $l$-th layer:

$$
\mathcal{G}^{(l)}\{\mathbf{u}, \mathbf{v}\}(f, f') := \delta(f - f')\frac{C_{R^{(l)}}}{n^{(l-1)}}\mathbb{E}\left[ \mathcal{F}\left( \mathfrak{S}^{(l-1)}\{\mathbf{u}\} \right)(f) \cdot \overline{\mathcal{F}\left( \mathfrak{S}^{(l-1)}\{\mathbf{v}\} \right)}(f') \right]. \tag{9}
$$

$$
\widetilde{\mathcal{G}^{(l)}}\{\mathbf{u}, \mathbf{v}\}(f, f') := \delta(f - f')\frac{C_{R^{(l)}}}{n^{(l-1)}}\sum_j \mathcal{F}\left( \mathfrak{S}^{(l-1)}\{\mathbf{u}\} \right)_j(f)\overline{\mathcal{F}\left( \mathfrak{S}^{(l-1)}\{\mathbf{v}\} \right)}_j(f'). \tag{10}
$$

The fluctuation of this metric is governed by the four-point connected correlator; see the Appendix B for a detailed derivation.

## 3.2 RUNNING OF COUPLINGS

In this subsection we derive recursion relations for the correlation functions—what, in field theory language, is the running of couplings. As the layer width tends to infinity, the pre-activation ensemble becomes Gaussian and the layer metric converges to a deterministic kernel $\mathcal{K}^{(l)}$ (see Appendix B). In this vanishing fluctuation regime, the next layer kernel can be expressed as a Gaussian expectation with covariance $\mathcal{K}^{(l)}$ inherited from the previous layer; equivalently, $\mathcal{K}^{(l+1)}$ is obtained by integrating over the Gaussian process defined by $\mathcal{K}^{(l)}$:

$$\mathcal{K}^{(l+1)}\{\mathbf{u}, \mathbf{v}\}(f, f') := \delta(f - f') C_{R^{(l+1)}}(f) \langle \mathcal{F}(\mathfrak{S}^{(l)}\{\mathbf{u}\}), \mathcal{F}(\mathfrak{S}^{(l)}\{\mathbf{v}\}) \rangle_{\mathcal{K}^{(l)}}(f, f').$$

Here, notation $\langle A, B \rangle_{\mathcal{K}}(f, f')$ means the expectation value of $A(f)\overline{B(f')}$ where each $A$ and $B$ are Gaussian random fields with kernel $\mathcal{K}$. We will denote functions $\mathcal{K}\{\mathbf{u}, \mathbf{u}\}$ simply as $\mathcal{K}\{\mathbf{u}\}$ and $\langle A, A \rangle_{\mathcal{K}}$ as $\|A\|_{\mathcal{K}}$. $\mathcal{K}\{\mathbf{u}\}(f)$ will mean the diagonal values $\mathcal{K}\{\mathbf{u}\}(f, f)$. Specifically, the first kernel is defined as in Eq. (11) and deeper layer kernels are calculated recursively as in Eq. (12).

$$\mathcal{K}^{(1)}\{\mathbf{u}, \mathbf{v}\}(f, f') := \frac{C_W}{n^{(1)}} \sum_j \hat{u}_j(f) \hat{v}_j(f'). \tag{11}$$

$$\mathcal{K}^{(l+1)}\{\mathbf{u}\}(f, f') := \delta(f - f') \frac{C_{R^{(l+1)}}}{n^{(l)}} g(\mathcal{K}^{(l)}),$$
$$g(\mathcal{K}) := \langle \mathcal{F}(\mathfrak{S}^{(l)}\{\mathbf{u}\}), \mathcal{F}(\mathfrak{S}^{(l)}\{\mathbf{u}\}) \rangle_{\mathcal{K}^{(l)}}(f, f'). \tag{12}$$

We analyze the layerwise amplification factor (susceptibility) of the kernel—i.e., the ratio of the kernel perturbation at layer $l+1$ to that at layer $l$—when an infinitesimal perturbation is injected into the previous layer. We consider both the parallel ($\chi_\parallel$) and the perpendicular ($\chi_\perp$) susceptibilities; formal definitions and derivations appear in Appendix C. Note that parallel and perpendicular susceptibilities are defined at different perturbative orders. Because the perturbation is orthogonal to the input in the perpendicular case, the leading non-vanishing contribution appears at second order in the small perturbation $\delta\boldsymbol{\eta}$. By contrast, a perturbation parallel to the input produces a non-zero contribution already at first order along parameter $\epsilon$. Each susceptibility is therefore defined in terms of the fluctuation that arises at its respective lowest non-trivial order. Accordingly, for local fluctuations to remain stable under perturbations perpendicular to the reference trajectory, the following condition must be satisfied:

$$\delta(f - f') C_{R^{(l+1)}} \left( \left\| \mathcal{F}\left(\mathfrak{S}'^{(l)}\{u_0\}\right) * \delta\boldsymbol{\eta} \right\|_{\mathcal{K}^{(l)}} \right)(f, f') = \mathcal{K}_\eta(f, f'). \tag{13}$$

Imposing that relation Eq. (13) exactly would make $\|\mathcal{F}(\mathfrak{S}'^{(l)}\{\mathbf{u_0}\})\|_{\mathcal{K}^{(l)}\{u_0\}}$ behave like a Dirac delta, which is too restrictive for most practical networks. Instead, we adopt a softer requirement: the integrated (layer-wise) sum of the kernel remains constant across layers, so the overall spectral energy is preserved while still allowing a non-trivial, spatially extended kernel shape.

$$\int \delta(f - f') C_{R^{(l+1)}} \|\mathcal{F}(\mathfrak{S}'^{(l)}\{\mathbf{u_0}\}) * \widehat{\delta\boldsymbol{\eta}}\|_{\mathcal{K}^{(l)}\{u_0\}} df df' = \int \mathcal{K}_\eta(f, f') df df'$$

$$\Rightarrow \int \delta(f - f') C_{R^{(l+1)}} \|\mathcal{F}(\mathfrak{S}'^{(l)}\{\mathbf{u_0}\})\|_{\mathcal{K}^{(l)}\{u_0\}}(f, f') df df' = 1$$

$$\Rightarrow \int C_{R^{(l+1)}}(f) \|\mathcal{F}(\mathfrak{S}'^{(l)}\{\mathbf{u_0}\})\|_{\mathcal{K}^{(l)}\{u_0\}}(f) df = 1.$$

For convenience we introduce the reduced perpendicular susceptibility

$$\tilde{\chi}_\perp(f, f') := \|\mathcal{F}(\mathfrak{S}'^{(l)}\{\mathbf{u_0}\})\|_{\mathcal{K}^{(l)}\{u_0\}}(f, f')$$

With this notation, the analysis of correlators and their recursion relations yields two depth-wise critical conditions:

$$\chi_\parallel \equiv 1, \quad \text{local condition (for parallel perturbation).}$$

$$\int C(f) \tilde{\chi}_\perp(f) df \equiv 1 \quad \text{global condition (for perpendicular perturbation).}$$

Here, $\tilde{\chi}_\perp$ is calculated at layer $l$ and $C(f)$ is $C_{R^{(l+1)}}(f)$.

## 4 RESULTS: THREE REPRESENTATIVE CASES

In this section we put the general framework of Section 3 to work by carrying out the calculation in three representative settings—(i) analytic activations, (ii) scale invariant architectures, and (iii) networks equipped with residual connections.

### 4.1 ANALYTIC ACTIVATIONS

First, we consider the case where the activation is an analytic function so that we can expand it and kernels perturbatively. We will frequently use the notation $\mathcal{H}^{(l)}(f) := \mathcal{K}^{(l)}(f)C_{R^{(l)}}(f)$; in the discrete case, $\mathcal{H}^{(l)}(n) := \mathcal{K}^{(l)}(n)C_{R^{(l)}}(n)$. Suppose the activation passes through the origin then it can be written as Eq. (14).

$$\sigma(x) := \sum_{n=1} \frac{\sigma_n}{n!}x^n \tag{14}$$

Following the analysis in Appendix D, we obtain the following expressions for the kernel recursion and the parallel/perpendicular susceptibilities:

**Theorem 1** With the analytic, origin passing activation specified in Eq. (14) and a Fourier Neural Operator defined by Eq. (4) under the initialization ensemble Eq. (5), the kernel and the susceptibilities are given by the following recursion relations:

$$\mathcal{K}^{(l+1)}(f, f') = \delta(f - f')C_{R^{(l+1)}}(f)\sum_{k=1}^{\infty} \frac{\sigma_k^2}{(n^{(l)})^{k-1}}$$

$$\sum_{n_{k,1}+\cdots+n_{k,l}=k} \frac{(n_{k,1})!\ldots(n_{k,l})!}{k!}(\mathcal{H}^{(l)})^{*n_{k,1}}\cdots(\mathcal{H}^{(l)})^{*n_{k,l}}(f),$$

$$\chi_\|(f, f') = \delta(f - f')C_{R^{(l+1)}}(f)\sum_{k=1}^{\infty} \frac{\sigma_k^2}{(n^{(l)})^{k-1}}$$

$$\sum_{n_{k,1}+\cdots+n_{k,l}=k} \frac{(n_{k,1})!\ldots(n_{k,l})!}{(k-1)!}\frac{(\mathcal{H}^{(l)})^{*n_{k,1}}\cdots(\mathcal{H}^{(l)})^{*n_{k,l}}(f)}{\mathcal{K}^{(l)}\{\mathbf{u_0}\}(f)},$$

$$\tilde{\chi}_\perp(f, f') = \delta(f - f')\sum_{k=1}^{\infty} \frac{\sigma_k^2}{(n^{(l)})^{k-1}}$$

$$\sum_{n_{k,1}+\cdots+n_{k,l}=k-1} \frac{(n_{k,1})!\ldots(n_{k,l})!}{(k-1)!}(\mathcal{H}^{(l)})^{*n_{k,1}}\cdots(\mathcal{H}^{(l)})^{*n_{k,l}}(f).$$

### 4.2 SCALE INVARIANT ACTIVATIONS

The scale invariant class of activations is defined as follows:

$$\sigma(z) := \begin{cases} \alpha z & \text{for } z \geq 0 \\ \beta z & \text{for } z < 0. \end{cases} \tag{15}$$

Following the analysis in Appendix E, we obtain the following expressions for the kernel recursion and the parallel/perpendicular susceptibilities:

**Theorem 2** With the scale invariant activation specified in Eq. (15) and a Fourier Neural Operator defined by Eq. (4) under the initialization ensemble Eq. (5), the kernel and the susceptibilities are

given by the following recursion relations:

$$\mathcal{K}^{(l+1)}(f, f') = (\alpha - \beta)^2 \delta(f - f') C_{R^{(l+1)}}(f)$$

$$\int \int \frac{V^{(l)}}{4\pi} \left( 2\sqrt{1 - (\rho^{(l)})^2} + \rho^{(l)}(\pi + 2\arcsin \rho^{(l)}) \right) e^{-ifx + if'x'} dx dx'$$

$$+ \alpha\beta\delta(f - f') C_{R^{(l+1)}}(f)\mathcal{H}^{(l)}(f),$$

$$\int \mathcal{K}^{(l+1)}(f, f') df df' = \frac{\alpha^2 + \beta^2}{2} \int \mathcal{H}^{(l)}(f) df$$

$$\chi_\parallel(f, f') = \frac{\mathcal{K}^{(l+1)}(f, f')}{\mathcal{K}^{(l)}(f, f')},$$

$$\tilde{\chi}_\perp(f, f') = \mathcal{F}_x \mathcal{F}_{x'} \left( \frac{(\alpha - \beta)^2}{4} \left( 1 + \frac{2}{\pi} \arcsin \rho^{(l)} \right) + \alpha\beta \right).$$

where $V^{(l)} = \int \mathcal{H}^{(l)}(f) df$, $V^{(l)}\rho^{(l)}(x, x') = \int \mathcal{H}^{(l)}(f) \cos(f(x - x')) df$. And each $\mathcal{F}_x, \mathcal{F}_{x'}$ denote the Fourier transform along each axis.

### 4.3 ARCHITECTURE WITH RESIDUAL CONNECTION

In practice, an FNO keeps only a subset of Fourier modes: after the FFT, all modes above a cutoff are truncated. With no activation (purely linear layers), both pre-activations and outputs stay strictly within this band. Once a nonlinearity is added, however, low frequencies couple to high frequency kernels—as shown in Sections 4.1–4.2 and confirmed empirically in Appendix H. For inputs of modest extent and large channel width, Theorem 1 implies that this coupling is weak. To retain the efficiency of truncation while still accumulating nonlinear interactions, we adopt a ResNet style update that adds each layer's pre-activation to its activation output. This helps preserve high frequency information with depth and improves fine scale processing. The rest of this section develops the corresponding theory for residual FNOs. We begin by defining a weighted residual connection for the kernel integration layer, modifying Eq. (4) accordingly.

$$\mathcal{Z}^{(l+1)}(u) := \mathcal{R}^{(l+1)}(\mathfrak{S}^{(l)}(u)) + \tilde{\gamma}\mathcal{Z}^{(l)}(u). \tag{16}$$

where $\tilde{\gamma}$ is a hyperparameter to be set. By extending the arguments of Theorem 1 to the ResNet architecture in Appendix F, we obtain the following result:

**Theorem 3** With the analytic, origin passing activation specified in Eq. (14) and a Fourier Neural Operator defined based on Eq. (4) and modification of Fourier layers by Eq. (16) under the initialization ensemble Eq. (5), the kernel and the susceptibilities are given by the following recursion relations. Here $\gamma = \tilde{\gamma}^2$:

$$\mathcal{K}^{(l+1)}\{\mathbf{u}\}(f, f') = \delta(f - f') \Big( (\sigma_1^2 C_{R^{(l+1)}}(f) + \gamma)\mathcal{K}^{(l)}\{\mathbf{u}\}(f)$$

$$+ C_{R^{(l+1)}}(f) \sum_{k=2}^{\infty} \frac{\sigma_k^2}{(n^{(l)})^{k-1}} \sum_{n_{k,1} + \cdots + n_{k,l} = k} \frac{(n_{k,1})! \ldots (n_{k,l})!}{k!} (\mathcal{H}^{(l)})^{*n_{k,1}} \cdots (\mathcal{H}^{(l)})^{*n_{k,l}}(f) \Big),$$

$$\chi_\parallel(f, f') = \delta(f - f') \Big( \gamma + C_{R^{(l+1)}}(f) \sum_{k=1}^{\infty} \frac{\sigma_k^2}{(n^{(l)})^{k-1}}$$

$$\sum_{n_{k,1} + \cdots + n_{k,l} = k} \frac{k(n_{k,1})! \ldots (n_{k,l})!}{(k-1)!} \frac{(\mathcal{H}^{(l)})^{*n_{k,1}} \cdots (\mathcal{H}^{(l)})^{*n_{k,l}}(f)}{\mathcal{K}^{(l)}\{\mathbf{u_0}\}(f)} \Big),$$

$$\tilde{\chi}_\perp(f, f') = \delta(f - f') \Big( \sum_{k=1}^{\infty} \frac{\sigma_k^2}{(n^{(l)})^{k-1}}$$

$$\sum_{n_{k,1} + \cdots + n_{k,l} = k-1} \frac{(n_{k,1})! \ldots (n_{k,l})!}{(k-1)!} (\mathcal{H}^{(l)})^{*n_{k,1}} \cdots (\mathcal{H}^{(l)})^{*n_{k,l}}(f) \Big) + (\gamma + 2\tilde{\gamma}\sigma_1)\delta(f)\delta(f').$$

Results for the compact (periodic) domain are provided in Appendix G.

## 5 Experimental Validations

In this section we empirically validate the theoretical findings from Section 4 and illustrate how they can be exploited in practice. We run three complementary sets of experiments. First, we verify qualitatively that a non-linear activation couples low and high frequencies, and then test the theory quantitatively by predicting the next layer kernel from the current one via Eqs. (24), (27) and (31) and comparing that prediction with measurements. Next, under fixed hyperparameters, we experimentally confirm that the susceptibilities in the Fourier layers behave as our theory predicts.

### 5.1 Kernel Evolution Prediction

In this subsection we start by examining how the kernel evolves under three different activations—quadratic, tanh, and ReLU—across a range of network depths and widths. We then study how the kernel changes when a tanh activation is used inside a ResNet-style FNO. Secondly, using the kernel recursion formula derived in Section 4, Appendix G, we feed the empirical kernel from layer $l$ into the recurrence to predict the kernel at layer $l + 1$ and compare that prediction with the measured kernel. To validate the frequency coupling and theoretical prediction over the truncation frequency ($k_{\max}$), we consider the reduced kernel $\tilde{\mathcal{K}}$ which is defined as follows:

$$\tilde{\mathcal{K}}^{(l+1)}\{\mathbf{u}, \mathbf{v}\}(f, f') := \left\langle \mathcal{F}(\mathfrak{S}^{(l)}\{\mathbf{u}\}), \mathcal{F}(\mathfrak{S}^{(l)}\{\mathbf{v}\}) \right\rangle_{\mathcal{K}^{(l)}}(f, f').$$

In the post-truncation band $f \in [17, 63]$ ($k_{max} = 16$), the measured kernels closely follow our theoretical curves across activation families (tanh, cubic, ReLU) and with/without a residual connection. With channel width fixed at 32 and depths 0,2 for tanh/cubic and 0,1,2,3 for ReLU/residual, the theory lies almost entirely within the $\pm 1$ s.d. band over 100 runs. Ablation tests that remove individual activation terms break this agreement, indicating that all relevant terms must be retained in the theory. The same behavior holds for non-constant spectra, e.g., $C_R(f) = \frac{2.5}{|f|}$. See Appendix H, Figs. 2 and 3 for full details.

### 5.2 Susceptibility Prediction

This section validates the parallel ($\chi_\parallel$) and perpendicular ($\chi_\perp$) susceptibilities predicted by our theory. We fix the architecture to truncation $k_{\max} = 32$ and channel width 32, with inputs drawn from the same Gaussian field as above, and read out the first Fourier layer. For the parallel tests we add a perturbation of size $\epsilon = 10^{-2}$ along the theory-specified data direction and compute $\chi_\parallel(f)$ via Eqs. (25), (29) and (32) averaging over 100 seeds. For the perpendicular tests we inject an orthogonal perturbation of size $\epsilon = 10^{-5}$, integrate the resulting power over all frequencies following Eqs. (26), (30) and (33) and again average over 100 runs. We evaluate four activation settings: quadratic, cubic, ReLU, and a ResNet variant with a cubic activation and residual gain $\gamma \in \{0.25, 0.75\}$. Except for the $C_R(f) \propto 1/|f|$ ablation, the weight-variance profile $C_R(f)$ is a step function that vanishes above $k_{\max}$; for the parallel tests we fix $C_R = 1$, while for the perpendicular tests we sweep $C_R \in [0.80, 1.20]$. Because non-ResNet models have zero kernel beyond $k_{\max}$, we show their curves only up to $k_{\max}$; the ResNet retains post-cutoff energy, so we plot it up to $2k_{\max}$. Figure 1 shows close agreement between theory and measurement: empirical susceptibilities lie within one standard deviation of the predicted curves across activations, perturbation types, and depths.

## 6 Conclusion

We analyze neural operators directly in function space, characterizing their frequency domain kernels and susceptibilities without resorting to discretization. The theory reveals that convolution in Fourier Neural Operators induces frequency coupling: even after spectral truncation, a reduced kernel retains energy at higher modes. The resulting behavior depends systematically on activation class—analytic versus non-analytic (with singular features)—and the predictions are borne out experimentally: the kernel evolution recursion matches measured kernels across depth, and both parallel and perpendicular susceptibilities agree with theory within empirical one-sigma bands.

These findings will provide practical criteria for selecting initialization hyperparameters and for anticipating spectral transfer as architectures deepen or widen. Promising next steps are to turn these criteria into concrete initialization/tuning procedures, broaden the analysis beyond the analytic,

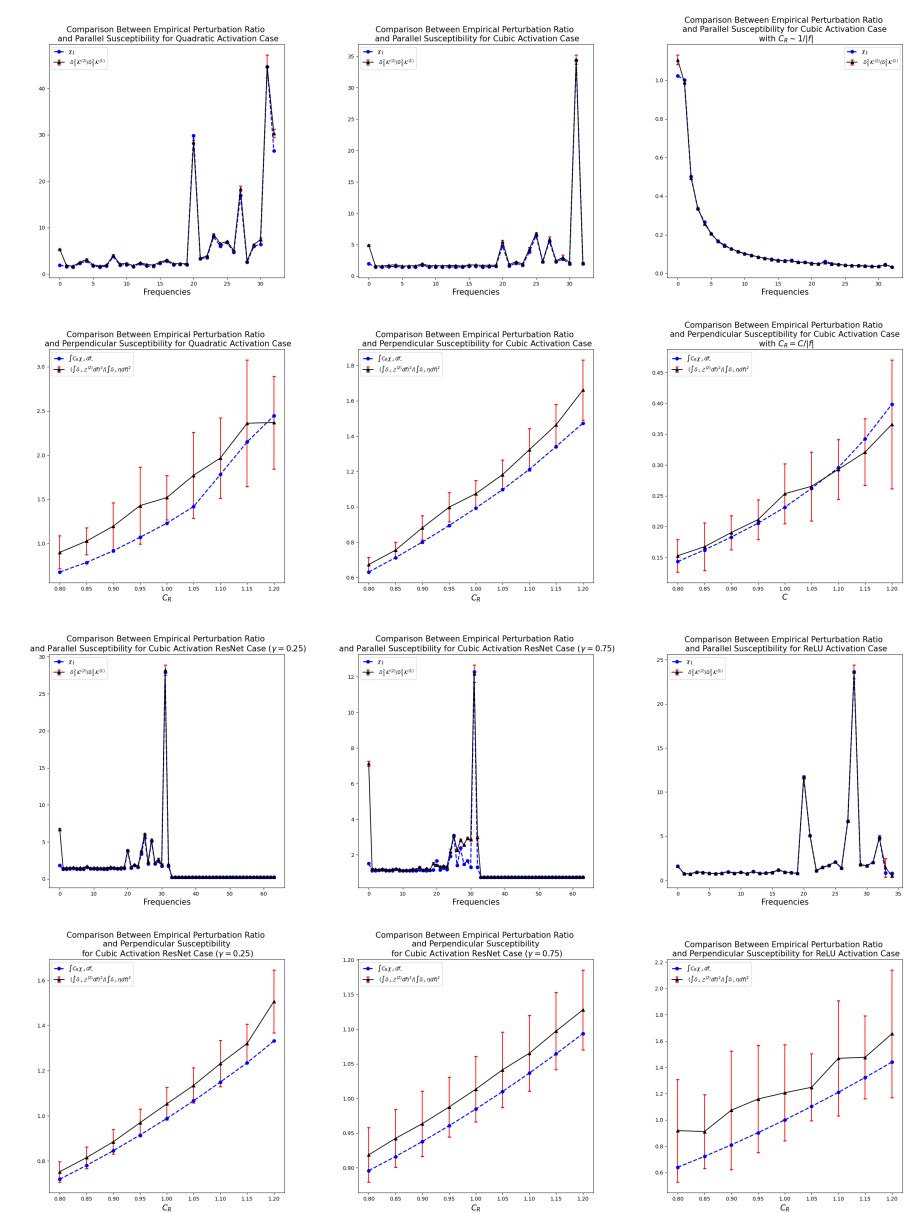

Figure 1: Empirical vs. theoretical susceptibilities (log scale; red error bars = $\pm 1$ s.d., $N = 100$) for width $n = 32$ and truncation $k_{\max} = 32$. Row 1: parallel susceptibility $\chi_{\parallel}$ for (col 1) quadratic with step $C_R$, (col 2) cubic with step $C_R$, (col 3) cubic with $C_R(f) \propto 1/|f|$ (truncated). Row 2: corresponding perpendicular susceptibility $\chi_{\perp}$. Row 3: $\chi_{\parallel}$ for cubic + residual with $\gamma = 0.25$ (col 1) and $\gamma = 0.75$ (col 2), and ReLU (col 3), all with step $C_R$. Row 4: corresponding $\chi_{\perp}$. Parallel runs use $\varepsilon_{\parallel} = 0.01$; perpendicular runs use $\varepsilon_{\perp} = 10^{-5}$. Markers = measurements; solid curves = theory.

scale invariant, and residual cases, track the ensemble distribution through training, and investigate higher-order correlations.

ACKNOWLEDGMENTS

Acknowledgments will be included in the camera-ready version.

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

## A  FCN Effective-Theory Background

We define the fully connected network (FCN), a basic architecture in deep learning models:

**Definition 3** The FCN is composed as follows:

$$z_i^{(1)} := \sum_{j=1}^{n^{(0)}} W_{ij}^{(1)} x_j + b_i^{(1)}$$

$$z_i^{(l+1)} := \sum_{j=1}^{n^{(l)}} W_{ij}^{(l+1)} \sigma(z_j^{(l)}) + b_i^{(l+1)}, \quad l = 1, \dots, L-1.$$

where $\sigma : \mathbb{R} \to \mathbb{R}$ is an activation function and $x$ is input-vector, $W^{(l)}$ and $b^{(l)}$ are weights and biases parameters. $z^{(l)}$ denotes the **pre-activation at the $l$-th layer**.

In deep learning architectures, the learning process involves optimizing the parameters $\{W^{(l)}, b^{(l)}\}_{l=1,\dots,L}$. The initial setting of these parameters, known as **initialization**, follows a specific probability distribution called the **initialization distribution**. For the statistical analysis of neural network pre-activations, we assume that the initialization distribution consists of independent and identically distributed (i.i.d.) Gaussian random variables. Specifically, we assume:

$$W_{ij}^{(l)} \sim N\left(0, \frac{C_W}{n^{(l-1)}}\right)$$

$$b_i^{(l)} \sim N(0, C_b).$$

Then, the statistics of first-layer pre-activation is also i.i.d Gaussian distribution with following covariances:

$$\mathbb{E}[z_{i_1;\alpha_1}^{(1)} z_{i_2;\alpha_2}^{(1)}] = \delta_{i_1 i_2} G_{\alpha_1 \alpha_2}^{(1)}$$

where the indices $\alpha_j$ are labels for input data, and $G_{\alpha_1 \alpha_2}^{(1)}$ is a metric for the first pre-activation, which contains all the information about the statistics. (since first pre-activation is Gaussian)

$$G_{\alpha_1 \alpha_2}^{(1)} = C_b^{(1)} + \frac{C_W^{(1)}}{n^{(0)}} \sum_{j}^{n^{(0)}} x_{j;\alpha_1} x_{j;\alpha_2}.$$

Now, the distribution of $l$-th layer pre-activation conditioned on $l-1$-th layer is also following Gaussian distribution:

$$\mathbb{E}[z_{i_1;\alpha_1}^{(l+1)} z_{i_2;\alpha_2}^{(l+1)} | z^{(l)}] = \delta_{i_1 i_2} \hat{G}_{\alpha_1 \alpha_2}^{(l)}$$

where $\hat{G}_{\alpha_1 \alpha_2}^{(l)}$ itself is a random variable, and defined as follows:

$$\hat{G}_{\alpha_1 \alpha_2}^{(l)} = C_b^{(l)} + \frac{C_W^{(l)}}{n^{(l-1)}} \sum_{j}^{n^{(l)}} z_{j;\alpha_1}^{(l-1)} z_{j;\alpha_2}^{(l-1)}.$$

Let the fluctuation of $\hat{G}_{\alpha_1 \alpha_2}^{(l)}$ around its mean $G_{\alpha_1 \alpha_2}^{(l)} := \mathbb{E}[\hat{G}_{\alpha_1 \alpha_2}^{(l)}]$ be $\Delta \hat{G}_{\alpha_1 \alpha_2}^{(l)} := \hat{G}_{\alpha_1 \alpha_2}^{(l)} - G_{\alpha_1 \alpha_2}^{(l)}$ then it can be shown that the variance of this fluctuation is related to four-point connected correlator and is $\mathcal{O}\left(\frac{1}{n^{(l-1)}}\right)$. Specifically, we define four-point vertex as follows:

$$V_{(\alpha_1 \alpha_2)(\alpha_3 \alpha_4)}^{(l)} := n^{(l-1)} \mathbb{E}[\Delta \hat{G}_{\alpha_1 \alpha_2}^{(l)} \Delta \hat{G}_{\alpha_3 \alpha_4}^{(l)}].$$

It can be easily checked that $2k$-point connected correlators is $\mathcal{O}\left(\frac{1}{n^{(l)k}}\right)$. So, for large enough widths, we can consider the neural network ensembles as a Gaussian processes. As widths go to infinity, the 2-point correlations go to some fixed kernel let denote this kernel as $K_{\alpha_1 \alpha_2}^{(l)}$. According to Roberts et al. (2022), for infinite width neural networks, the running of kernels and four-points

vertices are described as follows:

$$K_{\alpha_1\alpha_2}^{(l+1)} = C_b^{(l+1)} + C_W^{(l+1)}\langle\sigma_{\alpha_1}\sigma_{\alpha_2}\rangle_{K^{(l)}},$$

$$V_{(\alpha_1\alpha_2)(\alpha_3\alpha_4)}^{(l+1)} = (C_W^{(l+1)})^2[\langle\sigma_{\alpha_1}\sigma_{\alpha_2}\sigma_{\alpha_3}\sigma_{\alpha_4}\rangle_{K^{(l)}} - \langle\sigma_{\alpha_1}\sigma_{\alpha_2}\rangle_{K^{(l)}} - \langle\sigma_{\alpha_3}\sigma_{\alpha_4}\rangle_{K^{(l)}}]$$

$$+ \frac{n^{(l)}}{4n^{(l-1)}}(C_W^{(l+1)})^2 \sum_{\beta_1,\dots,\beta_4\in\mathcal{D}} V_{(l)}^{(\beta_1\beta_2)(\beta_3\beta_4)}\langle\sigma_{\alpha_1}\sigma_{\alpha_2}(z_{\beta_1}z_{\beta_2} - K_{\beta_1\beta_2}^{(l)})\rangle_{K^{(l)}} \quad (17)$$

$$\langle\sigma_{\alpha_3}\sigma_{\alpha_4}(z_{\beta_3}z_{\beta_4} - K_{\beta_3\beta_4}^{(l)})\rangle_{K^{(l)}} + O\left(\frac{1}{n^{(l)}}\right)$$

Let's now examine how the kernel changes under a small perturbation around a reference input $x_0$. Define two perturbed inputs $x_\pm = x_0 \pm \frac{1}{2}\delta x$, and let $z_+^{(l)}$ and $z_-^{(l)}$ denote their pre-activations at layer $l$. For this two-point dataset, the kernel admits an expansion of the form:

$$K_{\alpha\beta}^{(l)} = \begin{pmatrix} K_{++}^{(l)} & K_{+-}^{(l)} \\ K_{-+}^{(l)} & K_{--}^{(l)} \end{pmatrix} = K_{00}^{(l)}\begin{pmatrix} 1 & 1 \\ 1 & 1 \end{pmatrix} + K_{\parallel}^{(l)}\begin{pmatrix} 1 & 0 \\ 0 & -1 \end{pmatrix} + K_{\perp}^{(l)}\begin{pmatrix} 1 & -1 \\ -1 & 1 \end{pmatrix}.$$

where the coefficients are as follows:

$$K_{00}^{(l)} = \mathbb{E}\left[\frac{1}{n^{(l)}}\sum_i^{n^{(l)}} x_{i;0}^2\right],$$

$$K_{\parallel}^{(l)} = \frac{1}{2}\left(\mathbb{E}\left[\frac{1}{n^{(l)}}\sum_i^{n^{(l)}} x_{+;0}^2\right] - \mathbb{E}\left[\frac{1}{n^{(l)}}\sum_i^{n^{(l)}} x_{-;0}^2\right]\right),$$

$$K_{\perp}^{(l)} = \frac{1}{4}\left(\mathbb{E}\left[\frac{1}{n^{(l)}}\sum_i^{n^{(l)}} x_{+;0}^2\right] + \mathbb{E}\left[\frac{1}{n^{(l)}}\sum_i^{n^{(l)}} x_{-;0}^2\right] - 2\mathbb{E}\left[\frac{1}{n^{(l)}}\sum_i^{n^{(l)}} x_{+;0}x_{-;0}\right]\right).$$

As $\delta x \to 0$, the components $K_{\parallel}^{(l)}$ and $K_{\perp}^{(l)}$ each vanish, while the first term collapses to a degenerate matrix. In Roberts et al. (2022), through a careful eigenvalue expansion one finds that the two–point activation correlation under this perturbed kernel is given by:

$$\langle\sigma(z_\alpha)\sigma(z_\beta)\rangle_{K^{(l)}}$$
$$= \left[\sigma(z_0)\sigma(z_0)\rangle_{K_{00}^{(l)}}\right]\gamma_{\alpha\beta}^{[0]}$$
$$+ \left[\left(\frac{\delta K_{\parallel}^{(l)}}{K_{00}^{(l)}}\right)\langle z_0\sigma'(z_0)\sigma(z_0)\rangle_{K_{00}^{(l)}}\right]\gamma_{\alpha\beta}^{[1]}$$
$$+ \left[\delta\delta K_{\perp}^{(l)}\langle\sigma'(z_0)\sigma'(z_0)\rangle_{K_{00}^{(l)}} + \left(\frac{\delta K_{\parallel}^{(l)}}{2K_{00}^{(l)}}\right)^2\langle(z_0^2 - K_{00}^{(l)})\sigma'(z_0)\sigma'(z_0)\rangle_{K_{00}^{(l)}}\right]\gamma_{\alpha\beta}^{[2]}$$

From that result, one deduces the following recursion relations for the leading terms of $K_{\parallel}^{(l)}$ and $K_{\perp}^{(l)}$:

$$\delta K_{\parallel}^{(l+1)} = \frac{C_W}{K}\langle z\sigma'(z)\sigma(z)\rangle_K \delta K_{\parallel}^{(l)},$$

$$\delta\delta K_{\perp}^{(l+1)} = C_W\langle\sigma'(z)\sigma'(z)\rangle_K \delta\delta K_{\perp}^{(l)} + \frac{C_W}{4K^2}\langle\sigma'(z)\sigma'(z)(z^2 - K)\rangle_K(\delta K_{\parallel}^{(l)})^2.$$

When the perturbation is orthogonal to the data, the second term on the right-hand side of the equation for $\delta\delta K_{\perp}^{(l)}$ vanishes. Consequently, in both the parallel and perpendicular cases the two formulas above reduce to geometric-sequence recursions, and if their common ratios $\frac{C_W}{K}\langle z\sigma'(z)\sigma(z)\rangle_K$ and $C_W\langle\sigma'(z)\sigma'(z)\rangle_K$ both equal one, the volume occupied by the data distribution remains constant as it propagates through the layers.

# B  SCALING LAW AND FOUR-POINT VERTEX

**Large-Width Scaling of Connected Correlators** Eq. (18) presents the general form of an arbitrary $2k$-point connected correlator. Under mild regularity conditions, an inductive estimate shows that the correlator in layer $l$ obeys the scaling $\mathcal{O}(\frac{1}{n^s})$. If the activation function is analytic and vanishes at the origin, the leading contribution to the connected correlator of activations in layer $l-1$ obeys the same bound. In addition, the statistics of the pre-kernel integration stages match those of a fully connected network, as established in Section 2.1. Hence, when all widths are identical and large, $n = n_l = \cdots = n_0, n \gg 1$, the connected correlator in layer $l$ is suppressed to $\mathcal{O}(\frac{1}{n^{k-1}})$. In the infinite width limit, only the two-point connected correlator carries appreciable statistical weight; the analysis that follows therefore concentrates on the relationship between this kernel and the four-point vertex that governs its fluctuations. For conciseness the derivation in Eq. (18) is given for real-valued functions, but the extension to complex-valued fields is straightforward.

$$
\mathbb{E}\left[\mathcal{F}\left(\mathcal{Z}^{(l)}\{\mathbf{u}_1\}\right)_{i_1}(f_1)\ldots\mathcal{F}\left(\mathcal{Z}^{(l)}\{u_{2k}\}\right)_{i_{2k}}(f_{2k})\right]\Bigg|_{\text{conn}}
$$

$$
= \left(\frac{C_{R^{(l)}}}{n^{(l-1)}}\right)^k \delta_{i_1 i_2}\ldots\delta_{i_{2k-1}i_{2k}}\delta(f_1-f_2)\ldots\delta(f_{2k-1}-f_{2k})
$$

$$
\sum_{j_1,\ldots,j_{2k}}^{n^{(l-1)}} \delta_{j_1 j_2}\ldots\delta_{j_{2k-1}j_{2k}}\left\{\mathbb{E}\left[\mathcal{F}\left(\mathfrak{S}^{(l-1)}\{\mathbf{u}_1\}\right)_{j_1}(f_1)\ldots\mathcal{F}\left(\mathfrak{S}^{(l-1)}\{u_{2k}\}\right)_{j_{2k}}(f_{2k})\right] \right. \tag{18}
$$

$$
- \sum_{\text{all subdivisions of } (1,\ldots,2k)} \mathbb{E}\left[\mathcal{F}\left(\mathfrak{S}^{(l-1)}\{u_{\mu_{1,1}}\}\right)_{i_{\mu_{1,1}}}\ldots\mathcal{F}\left(\mathfrak{S}^{(l-1)}\{u_{\mu_{1,t_1}}\}\right)_{i_{\mu_{1,t_1}}}\right]|_{\text{conn}}\ldots
$$

$$
\left. \mathbb{E}\left[\mathcal{F}\left(\mathfrak{S}^{(l-1)}\{u_{\mu_{\nu,1}}\}\right)_{i_{\mu_{\nu,1}}}\ldots\mathcal{F}\left(\mathfrak{S}^{(l-1)}\{u_{\mu_{\nu,t_\nu}}\}\right)_{i_{\mu_{\nu,t_\nu}}}\right]|_{\text{conn}}\right\}
$$

**Four-point vertex** Henceforth, to obtain a well-defined four-point vertex, we factor out the $\delta(f-f')$ from the metric and work with the reduced metric. We define the fluctuation of metric $\Delta\widetilde{\mathcal{G}^{(l)}}$ as follows equation:

$$
\Delta\widetilde{\mathcal{G}^{(l)}}\{\mathbf{u},\mathbf{v}\}(f,f') := \widetilde{\mathcal{G}^{(l)}}(f,f') - \mathcal{G}^{(l)}(f,f')
$$

$$
= \frac{C_{R^{(l)}}}{n^{(l-1)}}\sum_j\left(\mathcal{F}\left(\mathfrak{S}^{(l-1)}\{\mathbf{u}\}\right)_j(f)\overline{\mathcal{F}\left(\mathfrak{S}^{(l-1)}\{\mathbf{v}\}\right)_j}(f')\right.
$$

$$
\left. - \mathbb{E}\left[\mathcal{F}\left(\mathfrak{S}^{(l-1)}\{\mathbf{u}\}\right)(f)\cdot\overline{\mathcal{F}\left(\mathfrak{S}^{(l-1)}\{\mathbf{v}\}\right)}(f')\right]\right)
$$

and define four-point vertex $\mathcal{V}^{(l)}\{(\mathbf{u}_1,\mathbf{u}_2),(\mathbf{u}_3,\mathbf{u}_4)\}$ which is scaled variance of fluctuation $\Delta\widetilde{\mathcal{G}^{(l)}}$. Using this quantity, we can calculate four-point correlators perturbatively. Eq. (19) shows the expansion of four-point vertex which is composed of four-point connected correlators of $(l-1)$-th layer activations.

$$
\frac{1}{n^{(l-1)}}\mathcal{V}^{(l)}\{(\mathbf{u}_1,\mathbf{u}_2),(\mathbf{u}_3,\mathbf{u}_4)\}(f_1,f_2,f_3,f_4) :=
$$

$$
\mathbb{E}\left[\Delta\widetilde{\mathcal{G}^{(l)}}\{\mathbf{u}_1,\mathbf{u}_2\}(f_1,f_2)\Delta\widetilde{\mathcal{G}^{(l)}}\{\mathbf{u}_3,\mathbf{u}_4\}(f_3,f_4)\right]
$$

$$
= C_{R^{(l)}}^2\left(\frac{1}{n^{(l-1)}}\right)^2\mathbb{E}\left[\sum_{j,k}\left(\mathcal{F}\left(\mathfrak{S}^{(l-1)}\{\mathbf{u}_1\}\right)_j(f_1)\mathcal{F}\left(\mathfrak{S}^{(l-1)}\{\mathbf{u}_2\}\right)_j(f_2)\right.\right.
$$

$$
\left. - \mathbb{E}\left[\mathcal{F}\left(\mathfrak{S}^{(l-1)}\{\mathbf{u}_1\}\right)(f_1)\cdot\mathcal{F}\left(\mathfrak{S}^{(l-1)}\{\mathbf{u}_2\}\right)(f_2)\right]\right)
$$

$$
\left.\left(\mathcal{F}\left(\mathfrak{S}^{(l-1)}\{\mathbf{u}_3\}\right)_k(f_3)\mathcal{F}\left(\mathfrak{S}^{(l-1)}\{\mathbf{u}_4\}\right)_k(f_4) - \mathbb{E}\left[\mathcal{F}\left(\mathfrak{S}^{(l-1)}\{\mathbf{u}_3\}\right)(f_3)\cdot\mathcal{F}\left(\mathfrak{S}^{(l-1)}\{\mathbf{u}_4\}\right)(f_4)\right]\right)\right]
$$

$$
\begin{aligned}
&= C_{R^{(l)}}^2 \Big(\frac{1}{n^{(l-1)}}\Big)^2 \sum_j \Big( \mathbb{E}\Big[ \mathcal{F}\Big(\mathfrak{S}^{(l-1)}\{\mathbf{u}_1\}\Big)_j(f_1) \mathcal{F}\Big(\mathfrak{S}^{(l-1)}\{\mathbf{u}_2\}\Big)_j(f_2) \\
&\quad \mathcal{F}\Big(\mathfrak{S}^{(l-1)}\{\mathbf{u}_3\}\Big)_j(f_3) \mathcal{F}\Big(\mathfrak{S}^{(l-1)}\{\mathbf{u}_4\}\Big)_j(f_4) \Big] - \mathbb{E}\Big[ \mathcal{F}\Big(\mathfrak{S}^{(l-1)}\{\mathbf{u}_1\}\Big)_j(f_1) \mathcal{F}\Big(\mathfrak{S}^{(l-1)}\{\mathbf{u}_2\}\Big)_j(f_2) \Big] \\
&\quad \mathbb{E}\Big[ \mathcal{F}\Big(\mathfrak{S}^{(l-1)}\{\mathbf{u}_3\}\Big)_j(f_3) \mathcal{F}\Big(\mathfrak{S}^{(l-1)}\{\mathbf{u}_4\}\Big)_j(f_4) \Big] \Big) \\
&\quad + C_{R^{(l)}}^2 \Big(\frac{1}{n^{(l-1)}}\Big)^2 \sum_{j \neq k} \Big( \mathbb{E}\Big[ \mathcal{F}\Big(\mathfrak{S}^{(l-1)}\{\mathbf{u}_1\}\Big)_j(f_1) \mathcal{F}\Big(\mathfrak{S}^{(l-1)}\{\mathbf{u}_2\}\Big)_j(f_2) \\
&\quad \mathcal{F}\Big(\mathfrak{S}^{(l-1)}\{\mathbf{u}_3\}\Big)_k(f_3) \mathcal{F}\Big(\mathfrak{S}^{(l-1)}\{\mathbf{u}_4\}\Big)_k(f_4) \Big] - \mathbb{E}\Big[ \mathcal{F}\Big(\mathfrak{S}^{(l-1)}\{\mathbf{u}_1\}\Big)_j(f_1) \mathcal{F}\Big(\mathfrak{S}^{(l-1)}\{\mathbf{u}_2\}\Big)_j(f_2) \Big] \\
&\quad \mathbb{E}\Big[ \mathcal{F}\Big(\mathfrak{S}^{(l-1)}\{\mathbf{u}_3\}\Big)_k(f_3) \mathcal{F}\Big(\mathfrak{S}^{(l-1)}\{\mathbf{u}_4\}\Big)_k(f_4) \Big] \Big)
\end{aligned}
\tag{19}
$$

Next, we describe the recursion for the four-point vertex. In Eq. (19), when all four indices coincide, the distribution reduces to one over a single scalar. Therefore, by computing each statistic similar to as in the nearly Gaussian action expansion of Roberts et al. (2022), we obtain the following expression as in Eq. (17):

$$
\begin{aligned}
&\frac{C_{R^{(l)}}^2}{n^{(l-1)}} \mathcal{V}^{(l)}\{(\mathbf{u}_{\alpha_1}, \mathbf{u}_{\alpha_2}), (\mathbf{u}_{\alpha_3}, \mathbf{u}_{\alpha_4})\} \\
&= \frac{1}{n^{(l)}} \Big[ \Big\langle \mathcal{F}\Big(\mathfrak{S}^{(l-1)}\Big)\{\mathbf{u}_{\alpha_1}\} \mathcal{F}\Big(\mathfrak{S}^{(l-1)}\Big)\{\mathbf{u}_{\alpha_2}\} \mathcal{F}\Big(\mathfrak{S}^{(l-1)}\Big)\{\mathbf{u}_{\alpha_3}\} \mathcal{F}\Big(\mathfrak{S}^{(l-1)}\Big)\{\mathbf{u}_{\alpha_4}\} \Big\rangle \Big] \\
&\quad - \Big\langle \mathcal{F}\Big(\mathfrak{S}^{(l-1)}\Big)\{\mathbf{u}_{\alpha_1}\} \mathcal{F}\Big(\mathfrak{S}^{(l-1)}\Big)\{\mathbf{u}_{\alpha_2}\} \Big\rangle \Big\langle \mathcal{F}\Big(\mathfrak{S}^{(l-1)}\Big)\{\mathbf{u}_{\alpha_3}\} \mathcal{F}\Big(\mathfrak{S}^{(l-1)}\Big)\{\mathbf{u}_{\alpha_4}\} \Big\rangle \Big] \\
&\quad + \frac{1}{4n^{(l-1)}} \sum_{\beta_1, \ldots, \beta_4 \in \mathcal{D}} \mathcal{V}_{(l-1)}^{-1}\{(\mathbf{u}_{\beta_1}, \mathbf{u}_{\beta_2}), (\mathbf{u}_{\beta_3}, \mathbf{u}_{\beta_4})\} \\
&\quad \Big\langle \Big( \mathcal{F}\Big(\mathfrak{S}^{(l-1)}\Big)\{\mathbf{u}_{\alpha_1}\} \mathcal{F}\Big(\mathfrak{S}^{(l-1)}\Big)\{\mathbf{u}_{\alpha_2}\} \Big( \mathcal{F}(\mathcal{Z}^{(l-1)}\{\mathbf{u}_{\beta_1}\}) \mathcal{F}(\mathcal{Z}^{(l-1)}\{\mathbf{u}_{\beta_2}\}) - \mathcal{G}^{(l-1)}\{\mathbf{u}_{\beta_1}, \mathbf{u}_{\beta_2}\} \Big) \Big) \Big\rangle_{\mathcal{G}^{(l-1)}} \\
&\quad \Big\langle \Big( \mathcal{F}\Big(\mathfrak{S}^{(l-1)}\Big)\{\mathbf{u}_{\alpha_3}\} \mathcal{F}\Big(\mathfrak{S}^{(l-1)}\Big)\{\mathbf{u}_{\alpha_4}\} \Big( \mathcal{F}(\mathcal{Z}^{(l-1)}\{\mathbf{u}_{\beta_1}\}) \mathcal{F}(\mathcal{Z}^{(l-1)}\{\mathbf{u}_{\beta_2}\}) - \mathcal{G}^{(l-1)}\{\mathbf{u}_{\beta_1}, \mathbf{u}_{\beta_2}\} \Big) \Big) \Big\rangle_{\mathcal{G}^{(l-1)}} \\
&\quad + \mathcal{O}\Big(\frac{1}{n^2}\Big)
\end{aligned}
\tag{20}
$$

We use angle brackets $\langle \cdot \rangle$ to indicate averaging with respect to the kernel (i.e., expectation under the kernel induced measure) in Eq. (20); this notation should not be confused with an inner product.

## C    SUSCEPTIBILITY RECURSIONS AND CRITICALITY CONDITIONS

As in Appendix A for FCNs, we derive the criticality condition by writing layerwise recursions for perturbations taken parallel and perpendicular to the data manifold. For the parallel case, fix a reference pre-activation $\mathbf{Z}_0$ and consider two nearby samples $\mathbf{Z}_{\pm}^{(l)} = (1 \pm \epsilon)\mathbf{Z}_0$, where $\epsilon > 0$ is the perturbation amplitude at layer $l$. Let $\mathbf{u}_0 := \mathcal{F}(\mathbf{Z}_0)$. Then the difference between $\big\|\mathcal{F}(\mathbf{Z}_+^{(l)})\big\|_{\mathcal{K}^{(l)}\{\mathbf{u}_0\}}$ and $\big\|\mathcal{F}(\mathbf{Z}_-^{(l)})\big\|_{\mathcal{K}^{(l)}\{\mathbf{u}_0\}}$ is $4\mathcal{K}^{(l)}\{\mathbf{u}_0\}\epsilon$. Performing a first–order Taylor expansion to evaluate the difference at layer $l+1$ yields:

$$
\begin{aligned}
&\|\mathcal{F}(\mathcal{Z}^{(l+1)}|_{\mathcal{Z}_0(1+\epsilon)})\|_{\mathcal{K}^{(l)}\{u_0\}} = C_{R^{(l+1)}} \|\mathcal{F}(\mathfrak{S}^{(l)}|_{\mathcal{Z}_0(1+\epsilon)})\|_{\mathcal{K}^{(l)}\{u_0\}} \\
&= C_{R^{(l+1)}} \|\mathcal{F}(\mathfrak{S}^{(l)}|_{\mathcal{Z}_0}) + \mathcal{F}(\mathcal{Z}_0 \mathfrak{S}'^{(l)}|_{\mathcal{Z}_0})\epsilon + \mathcal{O}(\epsilon^2)\|_{\mathcal{K}^{(l)}\{u_0\}} \\
&\Rightarrow \big\|\mathcal{F}(\mathbf{Z}_+^{(l+1)})\big\|_{\mathcal{K}^{(l)}\{\mathbf{u}_0\}} - \big\|\mathcal{F}(\mathbf{Z}_-^{(l+1)})\big\|_{\mathcal{K}^{(l)}\{\mathbf{u}_0\}} \\
&= 2C_{R^{(l+1)}} \Big( \langle \mathcal{F}(\mathcal{Z}_0 \mathfrak{S}'^{(l)}|_{\mathcal{Z}_0}), \mathcal{F}(\mathfrak{S}^{(l)}|_{\mathcal{Z}_0})\{\mathbf{u}_0\} \rangle_{\mathcal{K}^{(l)}\{u_0\}} + \langle \mathcal{F}(\mathfrak{S}^{(l)}|_{\mathcal{Z}_0}), \mathcal{F}(\mathcal{Z}_0 \mathfrak{S}'^{(l)}|_{\mathcal{Z}_0}) \rangle_{\mathcal{K}^{(l)}\{u_0\}} \Big) \epsilon.
\end{aligned}
$$

Therefore, for the difference in variances under parallel perturbations between the $l$-th and $(l+1)$-th layers to remain unchanged, the following equation must be satisfied:

$$\chi_\parallel(f, f') := \frac{\left\|\mathcal{F}(\mathbf{Z}_+^{(l+1)})\right\|_{\mathcal{K}^{(l)}\{\mathbf{u_0}\}} - \left\|\mathcal{F}(\mathbf{Z}_-^{(l+1)})\right\|_{\mathcal{K}^{(l)}\{\mathbf{u_0}\}}}{\left\|\mathcal{F}(\mathbf{Z}_+^{(l)})\right\|_{\mathcal{K}^{(l)}\{\mathbf{u_0}\}} - \left\|\mathcal{F}(\mathbf{Z}_-^{(l)})\right\|_{\mathcal{K}^{(l)}\{\mathbf{u_0}\}}}$$

$$= \frac{2C_{R^{(l+1)}}\left(\langle\mathcal{F}(\mathcal{Z}_0\mathfrak{S}'^{(l)}|_{\mathcal{Z}_0}), \mathcal{F}(\mathfrak{S}^{(l)}|_{\mathcal{Z}_0})\{\mathbf{u_0}\}\rangle_{\mathcal{K}^{(l)}\{u_0\}} + \langle\mathcal{F}(\mathfrak{S}^{(l)}|_{\mathcal{Z}_0}), \mathcal{F}(\mathcal{Z}_0\mathfrak{S}'^{(l)}|_{\mathcal{Z}_0})\rangle_{\mathcal{K}^{(l)}\{u_0\}}\right)\epsilon}{4\mathcal{K}^{(l)}\{\mathbf{u_0}\}\epsilon}$$

$$= \frac{C_{R^{(l+1)}}\left(\langle\mathcal{F}(\mathcal{Z}_0\mathfrak{S}'^{(l)}|_{\mathcal{Z}_0}), \mathcal{F}(\mathfrak{S}^{(l)}|_{\mathcal{Z}_0})\{\mathbf{u_0}\}\rangle_{\mathcal{K}^{(l)}\{u_0\}} + \langle\mathcal{F}(\mathfrak{S}^{(l)}|_{\mathcal{Z}_0}), \mathcal{F}(\mathcal{Z}_0\mathfrak{S}'^{(l)}|_{\mathcal{Z}_0})\rangle_{\mathcal{K}^{(l)}\{u_0\}}\right)(f, f')}{2\mathcal{K}^{(l)}\{\mathbf{u_0}\}(f, f')}$$

$$= 1.$$

Next, we consider perturbations in the perpendicular direction. In principle, the kernel on the two–point set $\{\mathcal{Z}_+, \mathcal{Z}_-\}$, with $\mathcal{Z}_+ = \mathcal{Z}_0 + \delta\boldsymbol{\eta}$ and $\mathcal{Z}_- = \mathcal{Z}_0 - \delta\boldsymbol{\eta}$, should be treated as a $2 \times 2$ matrix. However, under the orthogonality assumption $\delta\boldsymbol{\eta} \perp \mathcal{Z}_0$, all mixed terms proportional to $\delta\boldsymbol{\eta} \cdot \mathcal{Z}_0$ vanish, and the joint law of the norms factorizes as

$$\mathcal{P}(f) \propto \exp\left(-\frac{\|\mathcal{Z}_0\|^2(f)}{2\,\mathcal{K}_0(f)} - \frac{\|\delta\boldsymbol{\eta}\|^2(f)}{2\,\mathcal{K}_\eta(f)}\right).$$

Therefore, the variance of the difference $\mathfrak{S}^{(l)}\{u_+\} - \mathfrak{S}^{(l)}\{u_-\}$ admits the simple form below, where $\mathbf{u}_\pm := \mathcal{F}(\mathcal{Z}_\pm)$.

$$\frac{\|\mathfrak{S}^{(l)}\{u_+\} - \mathfrak{S}^{(l)}\{u_-\}\|_{\mathcal{K}^{(l)}}}{4}$$

$$= \frac{\mathcal{K}^{(l+1)}\{u_+\}(f) + \mathcal{K}^{(l+1)}\{u_-\}(f) - \mathcal{K}^{(l+1)}\{u_+, u_-\}(f) - \mathcal{K}^{(l+1)}\{u_-, u_+\}(f)}{4}$$

$$= \frac{\langle\mathcal{F}(\mathfrak{S}^{(l)}\{u_0\}) + \mathcal{F}(\mathfrak{S}'^{(l)}\{u_0\}\delta\boldsymbol{\eta}) + \mathcal{O}(\delta\boldsymbol{\eta}^2), \mathcal{F}(\mathfrak{S}^{(l)}\{u_0\}) + \mathcal{F}(\mathfrak{S}'^{(l)}\{u_0\}\delta\boldsymbol{\eta}) + \mathcal{O}(\delta\boldsymbol{\eta}^2)\rangle_{\mathcal{K}^{(l)}}}{4}$$

$$+ \frac{\langle\mathcal{F}(\mathfrak{S}^{(l)}\{u_0\}) - \mathcal{F}(\mathfrak{S}'^{(l)}\{u_0\}\delta\boldsymbol{\eta}) + \mathcal{O}(\delta\boldsymbol{\eta}^2), \mathcal{F}(\mathfrak{S}^{(l)}\{u_0\}) - \mathcal{F}(\mathfrak{S}'^{(l)}\{u_0\}\delta\boldsymbol{\eta}) + \mathcal{O}(\delta\boldsymbol{\eta}^2)\rangle_{\mathcal{K}^{(l)}}}{4}$$

$$- \frac{\langle\mathcal{F}(\mathfrak{S}^{(l)}\{u_0\}) - \mathcal{F}(\mathfrak{S}'^{(l)}\{u_0\}\delta\boldsymbol{\eta}) + \mathcal{O}(\delta\boldsymbol{\eta}^2), \mathcal{F}(\mathfrak{S}^{(l)}\{u_0\}) + \mathcal{F}(\mathfrak{S}'^{(l)}\{u_0\}\delta\boldsymbol{\eta}) + \mathcal{O}(\delta\boldsymbol{\eta}^2)\rangle_{\mathcal{K}^{(l)}}}{4}$$

$$- \frac{\langle\mathcal{F}(\mathfrak{S}^{(l)}\{u_0\}) + \mathcal{F}(\mathfrak{S}'^{(l)}\{u_0\}\delta\boldsymbol{\eta}) + \mathcal{O}(\delta\boldsymbol{\eta}^2), \mathcal{F}(\mathfrak{S}^{(l)}\{u_0\}) - \mathcal{F}(\mathfrak{S}'^{(l)}\{u_0\}\delta\boldsymbol{\eta}) + \mathcal{O}(\delta\boldsymbol{\eta}^2)\rangle_{\mathcal{K}^{(l)}}}{4}$$

$$= \left\|\mathcal{F}\left(\mathfrak{S}'^{(l)}\{u_0\}\right) * \widehat{\delta\boldsymbol{\eta}}\right\|_{\mathcal{K}^{(l)}} + \mathcal{O}(|\delta\boldsymbol{\eta}|^3).$$

$$(21)$$

Here, the kernel $\mathcal{K}^{(l)}$ in Eq. (21) is defined on the joint space that includes the distribution of $\delta\boldsymbol{\eta}$; Since the variance of $\mathbf{u}_+ - \mathbf{u}_-$ at the $l$-th layer is $4\mathcal{K}_\eta$ and the variance of the pre-activation at the $l+1$-th layer is derived in Eq. (21) the variance of the difference between the two data points will remain constant across layers only if the corresponding condition is satisfied:

$$\chi_\perp(f, f') := \frac{\|\mathcal{Z}^{(l+1)}|_{\mathcal{Z}_0+\delta\boldsymbol{\eta}} - \mathcal{Z}^{(l+1)}|_{\mathcal{Z}_0-\delta\boldsymbol{\eta}}\|_{\mathcal{K}^{(l)}}}{\|\mathcal{F}(\mathcal{Z}_0 + \delta\boldsymbol{\eta}) - \mathcal{F}(\mathcal{Z}_0 - \delta\boldsymbol{\eta})\|_{\mathcal{K}_\eta}}$$

$$= \delta(f - f')C_{R^{(l+1)}}\frac{\left\|\mathcal{F}\left(\mathfrak{S}'^{(l)}\{u_0\}\right) * \widehat{\delta\boldsymbol{\eta}}\right\|_{\mathcal{K}^{(l)}}(f, f')}{\mathcal{K}_\eta(f, f')} = 1.$$

# D   PROOFS FOR THEOREM 1

To handle the non-linearity introduced by an analytic activation in Fourier space, we first state the following lemma.

**Lemma 1** Let $\sigma : \mathbb{R} \to \mathbb{R}$ be an analytic function passing through the origin and $g \in L^1$ such that $\sigma(g) \in L^1$ then,

$$\text{For } \sigma(x) = \sum_{n=1}^{\infty} \frac{\sigma_n}{n!} x^n,$$

$$\mathcal{F}(\sigma(g))(f) = \sum_{n=1}^{\infty} \frac{\sigma_n}{n!} \hat{g}^{*n}(f).$$

where $\hat{g}^{*n}$ means $n$-times self-convolution of $\hat{g}$.

Then, the kernel can be written as follows:

$$\mathcal{K}^{(l+1)}\{\mathbf{u}\}(f) = C_{R^{(l+1)}}(f) \int \left\| \mathcal{F}\Big(\sigma(k^{(l)} * u)\Big) \right\|^2 (f) e^{-\int \frac{1}{2\mathcal{K}^{(l)}\{\mathbf{u}\}(f)} \hat{u}(f)^2 df} \mathcal{D}u$$

$$= \int \sum_i \frac{1}{n^{(l)}} \left\| \sigma_1 R_{ij}^{(l)}(f)\hat{u}_j(f) + \frac{\sigma_2}{2!}\Big((R_{ij}^{(l)}\hat{u}_j) * (R_{ij'}^{(l)}\hat{u}_{j'})\Big)(f) + \cdots \right\|^2 e^{-\int \frac{1}{2\mathcal{K}^{(l)}\{\mathbf{u}\}(f)} \hat{u}(f)^2 df} \mathcal{D}u$$

When the network output is expanded analytically and its Fourier transform is taken, the result can be written as a sum of terms in which a Gaussian random field is composed with itself multiple times. Evaluating those terms requires Lemma 2. The lemma is stated for complex, scalar-valued fields, but the vector-valued case follows immediately: simply apply the lemma componentwise, using the distributive property to handle each vector entry separately.

**Lemma 2** Suppose independent, mean-zero Gaussian random fields $\{R(f)\}_{f \in \mathbb{R}}, \{U(f)\}_{f \in \mathbb{R}}$ have following two-point correlators:

$$\mathbb{E}[R(f)\overline{R(f')}] = \delta(f - f')C(f),$$
$$\mathbb{E}[\text{Re}(R(f))\text{Im}(R(f'))] = 0,$$
$$\mathbb{E}[U(f)U(f')] = K(f, f').$$

then we have the following relations:

$$\mathbb{E}[(RU)^{*n}(f)(\overline{RU})^{*m}(f')]$$
$$= \delta_{n,m}(n!)^2 \delta(f - f')H^{*n}(f').$$

where $H(f) = C(f)K(f, f)$.

*Proof.*

$$\mathbb{E}[(RU)^{*n}(f)(\overline{RU})^{*m}(f')]$$

$$= \mathbb{E}[\int \delta(f - \sum_{k=1}^{n} f^{(k)}) \prod_{k=1}^{n}(RU)(f^{(k)})df^{(k)} \int \delta(f' - \sum_{k=1}^{m} f'^{(k)}) \prod_{k=1}^{m}(\overline{RU})(f'^{(k)})df^{(m)}]$$

$$= \int \delta(f - \sum_{k=1}^{n} f^{(k)})\delta(f' - \sum_{k=1}^{m} f'^{(k)})$$

$$\mathbb{E}[\prod_{k=1}^{n}\prod_{l=1}^{m} R(f^{(k)})\overline{R}(f^{(l)})]\mathbb{E}[\prod_{k=1}^{n}\prod_{l=1}^{m} U(f^{(k)})\overline{U}(f^{(l)})] \prod_{k=1}^{n}\prod_{l=1}^{n} df^{(k)}df'^{(l)}$$

if $n \neq m$, the expectation of $\prod_{l=1}^{m} R(f^{(k)})\overline{R}(f^{(l)})$ is zero since the wick contraction of each term contains expectation of $R^2$ or $\overline{R}^2$. So assuming $n = m$ then,

$$\mathbb{E}[(RU)^{*n}(f)(\overline{RU})^{*m}(f')]$$

$$= \delta_{n,m} \int \delta(f - \sum_{k=1}^{n} f^{(k)}) \delta(f' - \sum_{k=1}^{n} f'^{(k)})$$

$$\mathbb{E}[\prod_{k=1}^{n} \prod_{l=1}^{n} R(f^{(k)}) \overline{R}(f'^{(l)})] \mathbb{E}[\prod_{k=1}^{n} \prod_{l=1}^{n} U(f^{(k)}) \overline{U}(f'^{(l)})] \prod_{k=1}^{n} \prod_{l=1}^{n} df^{(k)} df'^{(l)}$$

$$= \delta_{n,m} \int \delta(f - \sum_{k=1}^{n} f^{(k)}) \delta(f' - \sum_{k=1}^{n} f'^{(k)}) (n!)^2$$

$$\Big( \prod_{k=1}^{n} \delta(f^{(k)} - f'^{(k)}) C(f^{(k)}) \Big) \Big( \prod_{k=1}^{n} K(f^{(k)}, f'^{(k)}) \Big) \prod_{k=1}^{n} df^{(k)} df'^{(k)}$$

$$= \delta_{n,m}(n!)^2 \int \delta(f - \sum_{k=1}^{n} f^{(k)}) \delta(f' - \sum_{k=1}^{n} f^{(k)})$$

$$\Big( \prod_{k=1}^{n} \delta(f^{(k)} - f'^{(k)}) C(f^{(k)}) \Big) \Big( \prod_{k=1}^{n} K(f^{(k)}, f'^{(k)}) \Big) \prod_{k=1}^{n} df^{(k)} df'^{(k)}$$

$$= \delta_{n,m}(n!)^2 \int \delta(f - \sum_{k=1}^{n} f^{(k)}) \delta(f' - \sum_{k=1}^{n} f^{(k)}) \Big( \prod_{k=1}^{n} H(f^{(k)}) \Big) \prod_{k=1}^{n} df^{(k)}$$

$$= \delta_{n,m}(n!)^2 \delta(f - f') H^{*n}(f').$$

$$\square$$

Using Lemma 2, we can now express the kernel of the next layer in terms of the current layer's kernel through the following expansion:

$$\mathcal{K}^{(l+1)}(f, f') = \sum_i \frac{C_{R^{(l+1)}}(f)}{n^{(l+1)}} \Big\| \Big( \sigma_1 R_{ij}^{(l)}(f) \hat{u}_j(f) + \frac{\sigma_2}{2!} \Big( (R_{ij}^{(l)} \hat{u}_j) * (R_{ij'}^{(l)} \hat{u}_{j'}) \Big)(f) + \cdots \Big) \Big\|_{\mathcal{K}^{(l)}}$$

$$= C_{R^{(l+1)}}(f) \sum_i \frac{1}{n^{(l)}} \sum_{k=1}^{\infty} \frac{\sigma_k^2}{(k!)^2} \|(R_{ij}^{(l)}(f) \hat{u}_j(f))^{*k}\|_{\mathcal{K}^{(l)}}$$

$$= \delta(f - f') C_{R^{(l+1)}}(f) \sum_{k=1}^{\infty} \frac{\sigma_k^2}{(n^{(l)})^{k-1}} \sum_{n_{k,1}+\cdots+n_{k,l}=k} \frac{(n_{k,1})! \dots (n_{k,l})!}{k!} (\mathcal{H}^{(l)})^{*n_{k,1}} \cdots (\mathcal{H}^{(l)})^{*n_{k,l}}(f)$$

In the second line, every term that cannot be paired with its complex conjugate drops out by Lemma 2. Combinatorially, at order $k$, the number of surviving monomials associated with a multiplicity vector $(n_{k,1}, \dots, n_{k,l})$ is $\frac{k!}{n_{k,1}! \dots n_{k,l}!}$. Applying Lemma 2 once more, each pair of identical factors contributes an additional factor $(n_{k,j}!)^2$. Collecting these factors for all indices produces the coefficients $\frac{(n_{k,1})! \dots (n_{k,l})!}{k!}$ which yields the expression shown in the third line. And for parallel susceptibility, we get the following:

$$\chi_{\|}(f, f') =$$

$$\frac{C_{R^{(l+1)}} \Big( \langle \mathcal{F}(\mathcal{Z}_0 \mathfrak{S}'^{(l)}|_{\mathcal{Z}_0}), \mathcal{F}(\mathfrak{S}^{(l)}|_{\mathcal{Z}_0}) \{\mathbf{u_0}\} \rangle_{\mathcal{K}^{(l)}\{u_0\}} + \langle \mathcal{F}(\mathfrak{S}^{(l)}|_{\mathcal{Z}_0}), \mathcal{F}(\mathcal{Z}_0 \mathfrak{S}'^{(l)}|_{\mathcal{Z}_0}) \rangle_{\mathcal{K}^{(l)}\{u_0\}} \Big)}{2\mathcal{K}^{(l)}\{\mathbf{u_0}\}}$$

$$= C_{R^{(l+1)}} \frac{\sum_i \frac{1}{n^{(l)}} \Big\langle \sum_{k=1} \frac{\sigma_k}{(k-1)!} \Big( R_{ij}^{(l)}(f) \hat{u}_i(f) \Big)^{*k}, \sum_{k=1} \frac{\sigma_k}{(k)!} \Big( R_{ij}^{(l)}(f) \hat{u}_i(f) \Big)^{*k} \Big\rangle_{\mathcal{K}^{(l)}}}{\mathcal{K}^{(l)}\{\mathbf{u_0}\}}$$

$$= C_{R^{(l+1)}} \sum_i \frac{1}{n^{(l)}} \sum_{k=1}^\infty \frac{\sigma_k^2}{(k-1)!k!} \frac{\langle (R_{ij}^{(l)}(f)\hat{u}_j(f))^{*k}, (R_{ij}^{(l)}(f)\hat{u}_j(f))^{*k} \rangle_{\mathcal{K}^{(l)}}}{\mathcal{K}^{(l)}\{\mathbf{u_0}\}}.$$

$$= \delta(f-f') C_{R^{(l+1)}} \sum_{k=1}^\infty \frac{\sigma_k^2}{(n^{(l)})^{k-1}} \sum_{n_{k,1}+\cdots+n_{k,l}=k} \frac{(n_{k,1})!\ldots(n_{k,l})!}{(k-1)!} \frac{(\mathcal{H}^{(l)})^{*n_{k,1}}\cdots(\mathcal{H}^{(l)})^{*n_{k,l}}(f)}{\mathcal{K}^{(l)}\{\mathbf{u_0}\}(f)}.$$

and for perpendicular susceptibility, we get the following:

$$\tilde{\chi}_\perp(f,f') = \|\mathcal{F}(\mathfrak{S}'^{(l)}\{\mathbf{u_0}\})\|_{\mathcal{K}^{(l)}\{u_0\}}(f,f')$$

$$= \sum_i \frac{1}{n^{(l)}} \|\sigma_1 + \sigma_2 R_{ij}^{(l)}(f)\hat{u}_j(f) + \frac{\sigma_3}{2!}\left((R_{ij}^{(l)}\hat{u}_j)*(R_{ij'}^{(l)}\hat{u}_{j'})\right)(f) + \cdots\|_{\mathcal{K}^{(l)}\{u_0\}}$$

$$= \delta(f-f') \sum_{k=1}^\infty \frac{\sigma_k^2}{(n^{(l)})^{k-1}} \sum_{n_{k,1}+\cdots+n_{k,l}=k-1} \frac{(n_{k,1})!\ldots(n_{k,l})!}{(k-1)!}(\mathcal{H}^{(l)})^{*n_{k,1}}\cdots(\mathcal{H}^{(l)})^{*n_{k,l}}(f).$$

# E  PROOFS FOR THEOREM 2

Since the sequence of functions $\sigma_n(x) = \frac{x}{2}(1 + \text{erf}(nx))$ converges to $\max(x,0)$, we have the following approximation:

**Lemma 3**  Let $\text{ReLU}(x) := \max(x,0)$ then we have the following convergent function sequence.

$$\text{ReLU}(x) \simeq \sigma_n(x) = \frac{1}{2}x + \frac{n}{\sqrt{\pi}}x^2 - \frac{n^3}{3\sqrt{\pi}}x^4 + \cdots$$

And to approximate the correlation of ReLU via $\sigma_n$'s, we need following lemma:

**Lemma 4**  Suppose $g_n$ converges pointwise to $g$ and $h \in L^1 \cap L^\infty$, satisfying following conditions:

- $g_n(x) \le C|x|$ for some constant $C > 0$.

then we have the following:

$$\mathcal{F}(g_n \circ h) \to \mathcal{F}(g \circ h) \quad \text{(uniformly)}$$

*Proof.*
$$|g_n \circ h(x)| \le C|h(x)|.$$

So, $\{g_n(h)\}$ is dominated by $C|h(x)|$. By dominated convergence theorem we get the following:

$$\lim_{n\to\infty} \int |g_n \circ h(x) - g \circ h(x)|dx = 0.$$

then, from the following inequality we get the uniform convergence of $g_n \circ h$ to $g(h)$:

$$|\mathcal{F}(g_n \circ h) - \mathcal{F}(g \circ h)| \le \int |(g_n \circ h - g \circ h)e^{-ifx}|dx = \int |g_n \circ h - g \circ h|dx = \|g_n \circ h - g \circ h\|_{L^1}.$$

$$\square$$

**Lemma 5**  Suppose mean zero Gaussian random fields $\{R(f)\}_{f \in \mathbb{R}}, \{U(f)\}_{f \in \mathbb{R}}$ have following two-point correlators:

$$\mathbb{E}[R(f)R(f')] = \delta(f-f')C(f),$$
$$\mathbb{E}[U(f)U(f')] = K(f,f').$$

then we have the following relations:

$$\mathbb{E}\left[\left\|\text{ReLU}\left(\int U dR\right)\right\|^2\right] = \frac{1}{2}\int C(f)K(f)df.$$

*Proof.* By Ito isometry we have the following identity:

$$\mathbb{E}\left[\left\|\left(\int U dR\right)\right\|^2\right] = \int C(f)K(f)df. \tag{22}$$

And since the expectation of $\int U dR$ is zero, only half of samples affects to the expectation of first equation. So by halving Eq. (22), we get the result. $\qquad\square$

To compute the covariance value of absolute of random variables, we need following lemmas from Heydenreich & Hofstad (2009), Vleck (1943).

**Lemma 6** For two Gaussian random variables $X, Y$ with means zero and correlation $\rho$, the expectation of $|X||Y|$ is formulated as follows:

$$\mathbb{E}[|X||Y|] = \frac{2}{\pi}(\sqrt{1-\rho^2} + \rho \arcsin \rho),$$

$$\mathbb{E}[1_{\{X>0\}}1_{\{Y>0\}}] = \frac{1}{4} + \frac{1}{2\pi}\arcsin \rho.$$

**Lemma 7** Suppose mean zero complex-valued Gaussian random fields $\{R(f)\}_{f\in\mathbb{R}}, \{U(f)\}_{f\in\mathbb{R}}$ have following two-point correlators:

$$\mathbb{E}[R(f)\overline{R}(f')] = \delta(f-f')C(f),$$
$$\mathbb{E}[U(f)] = m(f),$$
$$\mathbb{E}[U(f)\overline{U}(f')] = K(f,f').$$

let us define random fields $I(x), H(x)$ as follows:

$$I(x) := \text{ReLU}\left(\int R(f)U(f)e^{ifx}df\right),$$

$$H(x) := 1_{\{\int R(f)U(f)e^{ifx}df>0\}}.$$

then the mean and two-point correlators are calculated as follows:

$$\mathbb{E}[I(x)] = \frac{1}{\sqrt{2\pi}}V^{\frac{1}{2}} \quad \text{for all } x \in \mathbb{R},$$

$$\mathbb{E}[H(x)] = \frac{1}{2} \quad \text{for all } x \in \mathbb{R},$$

$$\mathbb{E}[I(x)\overline{I}(x')] = \frac{V}{4\pi}(2\sqrt{1-\rho^2} + \rho(\pi + 2\arcsin \rho)),$$

$$\mathbb{E}[H(x)\overline{H}(x')] = \frac{1}{4} + \frac{1}{2\pi}\arcsin \rho,$$

$$\mathbb{E}[\overline{R}(f')\overline{U}(f')\mathcal{F}(I(x))] = \frac{1}{2}\delta(f-f')K(f,f')C(f)$$

where

$$V = \int C(f)K(f,f)df,$$

$$V\rho(x,x') = \int C(f)K(f,f)\cos(f(x-x'))df.$$

*Proof.* Firstly, since for fixed $x \in \mathbb{R}$, $\int R(f)U(f)e^{ifx}df$ is a mean-zero Gaussian distribution with variance $V := \int C(f)K(f,f)df$. The mean of $I(x)$ which can be seen as a mean over Truncated Gaussian distribution is as follows:

$$\mathbb{E}[I(x)] = \sqrt{\frac{V}{2\pi}}.$$

and we can decompose ReLU activation into sum of $\frac{x}{2}$ and $\frac{|x|}{2}$. So, we get the following:

$$\mathbb{E}[I(x)\overline{I}(x')]$$

$$= \mathbb{E}\left[\frac{\left(\int R(f)U(f)e^{ifx}df + \left|\int R(f)U(f)e^{ifx}df\right|\right)\left(\int \overline{R}(f)\overline{U}(f)e^{-ifx'}df + \left|\int \overline{R}(f)\overline{U}(f)e^{-ifx'}df\right|\right)}{4}\right]$$

$$= \frac{1}{4}\left(\mathcal{F}_x^{-1}\mathcal{F}_{x'}^{-1}\left(\mathbb{E}[R(f)\overline{R}(f')]\mathbb{E}[U(f)U(f')]\right) + \mathbb{E}[R(f)\overline{R}(f')U(f)\overline{U}(f')]1_{\int R(f)U(f)e^{ifx}df>0}\right]$$

$$+ \mathbb{E}[R(f)\overline{R}(f')U(f)\overline{U}(f')]1_{\int R(f)U(f)e^{ifx'}df>0}\right) + \mathbb{E}[\left|\int R(f)U(f)e^{ifx}df\right|\left|\int \overline{R}(f)\overline{U}(f)e^{-ifx'}df\right|]\right)$$

$$= \frac{1}{4}\left(\mathcal{F}_x^{-1}\mathcal{F}_{x'}^{-1}\left(\delta(f-f')C(f)K(f,f')\right) + V\frac{2}{\pi}\left(\sqrt{1-\rho^2} + \rho\arcsin\rho\right)\right)$$

$$= \frac{V}{4\pi}\left(2\sqrt{1-\rho^2} + \rho(\pi + 2\arcsin\rho)\right).$$

It is clear that $\mathbb{E}[H(x)] = \frac{1}{2}$ and the result for $\mathbb{E}[H(x)\overline{H}(x')]$ follows directly from the second equation of Lemma 6. Finally for the last equation, we have the following equation:

$$\mathbb{E}[\overline{R}(f')\overline{U}(f')\mathcal{F}(I(x))]$$
$$= \mathcal{F}(\mathbb{E}[\overline{R}(f')\overline{(U)}(f')I(x)]).$$

since $\int R(f)U(f)e^{ifx}df$ is mean-zero the second line of following equations holds,

$$\mathcal{F}(\mathbb{E}[\overline{R}(f')\overline{U}(f')I(x)]) = \mathcal{F}(\mathbb{E}[\int \overline{R}(f')R(f)\overline{U}(f')U(f)e^{ifx}1_{I(x)>0}])$$

$$= \mathcal{F}\frac{1}{2}\mathbb{E}[\mathcal{F}^{-1}(\overline{R}(f')R(f)\overline{U}(f')U(f))]$$

$$= \frac{1}{2}\mathcal{F}\mathcal{F}^{-1}\mathbb{E}[\overline{R}(f')R(f)\overline{U}(f')U(f)]$$

$$= \frac{1}{2}\delta(f-f')K(f,f')C(f).$$

$$\qquad\qquad\qquad\qquad\qquad\qquad\qquad\qquad\qquad\qquad\qquad\qquad \square$$

Now, consider the Fourier transform $\mathcal{F}(\mathfrak{S}^{(l)}\{\mathbf{u}\})$ in the case of scale invariant activation. We have

$$\mathcal{F}(\mathfrak{S}^{(l)}\{\mathbf{u}\}) = (\alpha - \beta)\int \text{ReLU}(\int R_{ij}^{(l)}\hat{u}_j e^{ifx}df)e^{-ifx}dx + \beta(R_{ij}^{(l)}\hat{u}_j).$$

then

$$\mathbb{E}\left[\mathcal{F}(\mathfrak{S}^{(l)}\{\mathbf{u}\})\overline{\mathcal{F}(\mathfrak{S}^{(l)}\{\mathbf{u}\})}\right](f,f')$$

$$\Rightarrow \mathbb{E}\left[(\alpha-\beta)^2\int \text{ReLU}(\int R_{ij}^{(l)}\hat{u}_j e^{ifx}df)e^{-ifx}dx \int \overline{\text{ReLU}(\int R_{ij}^{(l)}\hat{u}_j e^{ifx}df)}e^{if'x}dx\right.$$

$$+ (\alpha-\beta)\overline{\beta(R_{ij}^{(l)}\hat{u}_j)}\int \text{ReLU}(\int R_{ij}^{(l)}\hat{u}_j e^{ifx}df)e^{-ifx}dx$$

$$+ (\alpha-\beta)\beta(R_{ij}^{(l)}\hat{u}_j)\int \overline{\text{ReLU}(\int R_{ij}^{(l)}\hat{u}_j e^{ifx}df)}e^{-if'x'}dx' + \beta^2(R_{ij}^{(l)}\hat{u}_j)\overline{(R_{ij}^{(l)}\hat{u}_j)}\right]$$

$$= (\alpha-\beta)^2\int\int\left(\frac{V}{4\pi}(2\sqrt{1-\rho^2} + \rho(\pi + 2\arcsin\rho))\right)e^{-ifx+if'x'}dxdx'$$

$$(\alpha-\beta)\beta\delta(f-f')C(f)\mathcal{K}(f,f') + \beta^2\delta(f-f')C(f)\mathcal{K}(f,f').$$

$$(23)$$

And from the fact that $\int(f*g)(x)dx = (\int f(x)dx)(\int g(x)dx)$ we can calculate the total integration of kernel over the two-point frequency domain by transforming the convoluted terms into geometric

terms:

$$\int\int \mathbb{E}\Big[\mathcal{F}(\mathfrak{S}^{(l)}\{\mathbf{u}\})\overline{\mathcal{F}(\mathfrak{S}^{(l)}\{\mathbf{u}\})}\Big](f,f')df\,df'$$

$$\simeq \mathbb{E}\Big[(\alpha-\beta)^2\sigma_n\Big(\int \hat{u}_j R_{ij}^{(l)}(df)\Big)\overline{\sigma_n\Big(\int \hat{u}_j R_{ij}^{(l)}(df')\Big)} + (\alpha-\beta)\beta\overline{\int \hat{u}_j R_{ij}^{(l)}(df')}\sigma_n\Big(\int \hat{u}_j R_{ij}^{(l)}(df)\Big)$$

$$+ (\alpha-\beta)\beta\int \hat{u}_j R_{ij}^{(l)}(df)\overline{\sigma_n\Big(\int \hat{u}_j R_{ij}^{(l)}(df')\Big)} + \beta^2 \int R_{ij}^{(l)}(f)\overline{R_{ij'}^{(l)}}(f')\hat{u}_j(f)\hat{u}_{j'}(f')df\,df'\Big]$$

$$\Rightarrow \int \langle \mathcal{F}\mathfrak{S}^{(l)}\{\mathbf{u_0}\}, \mathcal{F}\mathfrak{S}^{(l)}\{\mathbf{u_0}\}\rangle_{\mathcal{K}^{(l)}\{u_0,u_0\}}(f,f')df\,df'$$

$$= \frac{\alpha^2+\beta^2}{2}\int \mathcal{H}^{(l)}(f)df.$$

From Eq. (23), if $\alpha \neq \beta$, the first term survives which mixes all the components in positional domain. Then this term makes the kernel not to vanish over the truncated frequencies. And now we consider the parallel susceptibility. For that we first consider the following term:

$$\langle \mathcal{F}(\mathcal{Z}_{\mathbf{0}}\mathfrak{S}'^{(l)}|_{\mathcal{Z}_0}), \mathcal{F}(\mathfrak{S}^{(l)}|_{\mathcal{Z}_0})\{\mathbf{u_0}\}\rangle_{\mathcal{K}^{(l)}\{u_0\}}(f,f')$$

$$= \mathcal{F}_x\mathcal{F}_{x'}\Big(\langle \mathcal{Z}_{\mathbf{0}}\mathfrak{S}'^{(l)}|_{\mathcal{Z}_0}, \mathfrak{S}^{(l)}|_{\mathcal{Z}_0}\rangle_{\mathcal{K}^{(l)}\{u_0\}}\Big)(f,f')$$

$$= \mathcal{F}_x\mathcal{F}_{x'}\Big(\langle \mathcal{Z}_0 H(\mathcal{Z}_0)(x), \mathfrak{S}^{(l)}|_{\mathcal{Z}_0}(x')\rangle\Big)(f,f')$$

$$= \mathcal{F}_x\mathcal{F}_{x'}\Big(\langle \mathfrak{S}^{(l)}|_{\mathcal{Z}_0}(x), \mathfrak{S}^{(l)}|_{\mathcal{Z}_0}(x')\rangle\Big)(f,f').$$

So, for parallel susceptibility we get the following simple result:

$$\chi_\parallel(f,f')$$

$$= \frac{C_{R^{(l+1)}}\Big(\langle \mathcal{F}(\mathcal{Z}_{\mathbf{0}}\mathfrak{S}'^{(l)}|_{\mathcal{Z}_0}), \mathcal{F}(\mathfrak{S}^{(l)}|_{\mathcal{Z}_0})\{\mathbf{u_0}\}\rangle_{\mathcal{K}^{(l)}\{u_0\}} + \langle \mathcal{F}(\mathfrak{S}^{(l)}|_{\mathcal{Z}_0}), \mathcal{F}(\mathcal{Z}_{\mathbf{0}}\mathfrak{S}'^{(l)}|_{\mathcal{Z}_0})\rangle_{\mathcal{K}^{(l)}\{u_0\}}\Big)}{2\mathcal{K}^{(l)}\{\mathbf{u_0}\}}$$

$$= \frac{\mathcal{K}^{(l+1)}(f,f')}{\mathcal{K}^{(l)}(f,f')}.$$

And for the reduced perpendicular susceptibility:

$$\tilde{\chi}_\perp(f,f') = \langle \mathcal{F}(\mathfrak{S}'^{(l)}\{\mathbf{u_0}\}), \mathcal{F}(\mathfrak{S}'^{(l)}\{\mathbf{u_0}\})\rangle_{\mathcal{K}^{(l)}\{u_0,u_0\}}$$

$$= \mathcal{F}_x\mathcal{F}_{x'}\Big(\Big\|\Big((\alpha-\beta)H(x)+\beta\Big)\Big\|_{\mathcal{K}^{(l)}\{u_0,u_0\}}\Big)$$

$$= \mathcal{F}_x\mathcal{F}_{x'}\Big(\frac{(\alpha-\beta)^2}{4}\Big(1+\frac{2}{\pi}\arcsin\rho\Big) + \alpha\beta\Big).$$

## F   DERIVATION OF THEOREM 3

The recursive kernel for residual connection architecture is modified as follows:

$$\mathcal{K}^{(l+1)}\{\mathbf{u},\mathbf{v}\}(f) := \langle \mathcal{F}(\mathcal{R}^{(l+1)}(\mathfrak{S}^{(l)}\{\mathbf{u}\}) + \tilde{\gamma}\mathcal{Z}^{(l)}\{\mathbf{u}\}), \mathcal{F}(\mathcal{R}^{(l+1)}(\mathfrak{S}^{(l)}\{\mathbf{v}\}) + \tilde{\gamma}\mathcal{Z}^{(l)}\{\mathbf{v}\})\rangle_{\mathcal{K}^{(l)}}$$

Then, for single input data, we get the following recursive formula:

$$\mathcal{K}^{(l+1)}\{\mathbf{u}\}(f) := C_{R^{(l+1)}}\|\mathcal{F}(\mathfrak{S}^{(l)}\{\mathbf{u}\})\|_{\mathcal{K}^{(l)}} + \tilde{\gamma}\langle \mathcal{F}(\mathcal{Z}^{(l)}\{\mathbf{u}\}), \mathcal{F}(\mathcal{R}^{(l+1)}(\mathfrak{S}^{(l)}\{\mathbf{u}\}))\rangle_{\mathcal{K}^{(l)}}$$

$$+ \tilde{\gamma}\langle \mathcal{F}(\mathcal{R}^{(l+1)}(\mathfrak{S}^{(l)}\{\mathbf{u}\})), \mathcal{F}(\mathcal{Z}^{(l)}\{\mathbf{u}\})\rangle_{\mathcal{K}^{(l)}} + \tilde{\gamma}^2\|\mathcal{F}(\mathcal{Z}^{(l)}\{\mathbf{u}\})\|_{\mathcal{K}^{(l)}}$$

then for analytic activation case, we have the following kernel recursive formula:

$$C_{R^{(l+1)}}\|\mathcal{F}(\mathfrak{S}^{(l)}\{\mathbf{u}\})\|_{\mathcal{K}^{(l)}} + \tilde{\gamma}\langle \mathcal{F}(\mathcal{Z}^{(l)}\{\mathbf{u}\}), \mathcal{F}(\mathcal{R}^{(l+1)}(\mathfrak{S}^{(l)}\{\mathbf{u}\}))\rangle_{\mathcal{K}^{(l)}}$$

$$+ \tilde{\gamma}\langle \mathcal{F}(\mathcal{R}^{(l+1)}(\mathfrak{S}^{(l)}\{\mathbf{u}\})), \mathcal{F}(\mathcal{Z}^{(l)}\{\mathbf{u}\})\rangle_{\mathcal{K}^{(l)}} + \tilde{\gamma}^2\|\mathcal{F}(\mathcal{Z}^{(l)}\{\mathbf{u}\})\|_{\mathcal{K}^{(l)}}$$

$$= \delta(f-f')\Big(\gamma\mathcal{K}^{(l)}\{\mathbf{u}\}(f)$$

$$+ C_{R^{(l+1)}}(f)\sum_{k=1}^{\infty}\frac{\sigma_k^2}{(n^{(l)})^{k-1}}\sum_{n_{k,1}+\cdots+n_{k,l}=k}\frac{k(n_{k,1})!\ldots(n_{k,l})!}{(k-1)!}(\mathcal{H}^{(l)})^{*n_{k,1}}\cdots(\mathcal{H}^{(l)})^{*n_{k,l}}(f)\Big).$$

$$\Rightarrow \mathcal{K}^{(l+1)}\{\mathbf{u}\}(f, f') = \delta(f - f')\Big((\sigma_1^2 C_{R^{(l+1)}}(f) + \gamma)\mathcal{K}^{(l)}\{\mathbf{u}\}(f)$$

$$+ \sum_{k=2}^{\infty} \frac{\sigma_k^2}{(n^{(l)})^{k-1}} \sum_{n_{k,1}+\cdots+n_{k,l}=k} \frac{(n_{k,1})!\ldots(n_{k,l})!}{k!} (\mathcal{H}^{(l)})^{*n_{k,1}} \cdots (\mathcal{H}^{(l)})^{*n_{k,l}}(f)\Big).$$

where we set $\gamma = \tilde{\gamma}^2$. For the susceptibilities, we proceed analogously; the parallel case is omitted for brevity:

$$\langle \mathcal{F}(\mathfrak{S}'^{(l)}\{\mathbf{u_0}\} + \tilde{\gamma}\mathbf{1}), \mathcal{F}(\mathfrak{S}'^{(l)}\{\mathbf{u_0}\} + \tilde{\gamma}\mathbf{1})\rangle_{\mathcal{K}^{(l)}\{u_0, u_0\}}$$

$$= \langle \mathcal{F}(\mathfrak{S}'^{(l)}\{\mathbf{u_0}\}), \mathcal{F}(\mathfrak{S}'^{(l)}\{\mathbf{u_0}\})\rangle_{\mathcal{K}^{(l)}\{u_0, u_0\}} + 2\langle \mathcal{F}(\mathfrak{S}'^{(l)}\{\mathbf{u_0}\}), \tilde{\gamma}\mathbf{1}\rangle_{\mathcal{K}^{(l)}\{u_0, u_0\}} + \gamma\delta(f)\delta(f')$$

$$= \delta(f - f')\Big(\sum_{k=1}^{\infty} \frac{\sigma_k^2}{(n^{(l)})^{k-1}} \sum_{n_{k,1}+\cdots+n_{k,l}=k-1} \frac{(n_{k,1})!\ldots(n_{k,l})!}{(k-1)!} (\mathcal{H}^{(l)})^{*n_{k,1}} \cdots (\mathcal{H}^{(l)})^{*n_{k,l}}(f)\Big)$$

$$+ 2\tilde{\gamma}\sigma_1\delta(f)\delta(f') + \gamma\delta(f)\delta(f').$$

Here $\mathbf{1} = (1, \ldots, 1)^T \in \mathbb{R}^{n^{(l)}}$.

# G   ON THE COMPACT PERIODIC DOMAIN

As noted above, when the domain is the entire real line, the formula in Lemma 2 yields a $\delta(0)$ term—i.e., a divergent value. In practice, however, we implement the FNO on a finite periodic domain, and in this setting the $\delta(0)$ in equations in previous sections equals to $\frac{1}{L}$ where $L$ is size of the domain, so the divergence disappears. Consequently, the kernel evolution equations and criticality conditions in Sections 4.1, 4.2, and 4.3 can be rewritten as follows:

**Corollary 1** With the analytic, origin passing activation specified in Eq. (14) and a Fourier Neural Operator defined by Eq. (4) under the initialization ensemble Eq. (5), the kernel and the susceptibilities are given by the following recursion relations:

$$\mathcal{K}^{(l+1)}(m, n) = \frac{\delta_{m,n}C_{R^{(l+1)}}(n)}{L} \sum_{k=1}^{\infty} \frac{\sigma_k^2}{(n^{(l)})^{k-1}} \sum_{n_{k,1}+\cdots+n_{k,l}=k} \frac{(n_{k,1})!\ldots(n_{k,l})!}{k!} (\mathcal{H}^{(l)})^{*n_{k,1}} \cdots (\mathcal{H}^{(l)})^{*n_{k,l}}(n)$$
$$(24)$$

$$\chi_{\parallel}(m, n) = \frac{\delta_{m,n}}{L} C_{R^{(l+1)}}(n) \sum_{k=1}^{\infty} \frac{\sigma_k^2}{(n^{(l)})^{k-1}} \sum_{n_{k,1}+\cdots+n_{k,l}=k} \frac{(n_{k,1})!\ldots(n_{k,l})!}{(k-1)!} \frac{(\mathcal{H}^{(l)})^{*n_{k,1}} \cdots (\mathcal{H}^{(l)})^{*n_{k,l}}(n)}{\mathcal{K}^{(l)}\{\mathbf{u_0}\}(n)},$$
$$(25)$$

$$\tilde{\chi}_{\perp}(m, n) = \delta_{m,n} \sum_{k=1}^{\infty} \frac{\sigma_k^2}{(n^{(l)})^{k-1}} \sum_{n_{k,1}+\cdots+n_{k,l}=k-1} \frac{(n_{k,1})!\ldots(n_{k,l})!}{(k-1)!} (\mathcal{H}^{(l)})^{*n_{k,1}} \cdots (\mathcal{H}^{(l)})^{*n_{k,l}}(n). \quad (26)$$

**Corollary 2** With the scale invariant activation specified in Eq. (15) and a Fourier Neural Operator defined by Eq. (4) under the initialization ensemble Eq. (5), the kernel and the susceptibilities are given by the following recursion relations:

$$\mathcal{K}^{(l+1)}(m, n) = (\alpha - \beta)^2 \frac{\delta_{m,n}C_{R^{(l+1)}}(n)}{L}$$
$$\int \frac{V^{(l)}}{4\pi}\Big(2\sqrt{1 - (\rho^{(l)})^2} + \rho^{(l)}(\pi + 2\arcsin\rho^{(l)})\Big)e^{-imx+inx'}dxdx' \quad (27)$$
$$+ \alpha\beta\frac{\delta_{m,n}C_{R^{(l+1)}}(n)}{L}\mathcal{H}^{(l)}(n),$$

$$\sum_n \mathcal{K}^{(l+1)}(n) = \frac{\alpha^2 + \beta^2}{2}\sum_n \mathcal{H}^{(l)}(n) \quad (28)$$

$$\chi_{\parallel}(m, n) = \frac{\mathcal{K}^{(l+1)}(m, n)}{\mathcal{K}^{(l)}(m, n)}, \quad (29)$$

$$\tilde{\chi}_\perp(m,n) = \int \left( \frac{(\alpha-\beta)^2}{4} \left(1 + \frac{2}{\pi} \arcsin \rho^{(l)}\right) + \alpha\beta \right) e^{-imx+inx'} dx dx'. \tag{30}$$

where $V^{(l)} = \sum_n \mathcal{K}^{(l)}(n)$, $V^{(l)} \rho^{(l)}(x,x') = \sum_n \mathcal{H}^{(l)}(n) \cos(n(x-x'))$.

**Corollary 3** With the analytic, origin passing activation specified in Eq. (14) and a Fourier Neural Operator defined based on Eq. (4) and modification of Fourier layers by Eq. (16) under the initialization ensemble Eq. (5), the kernel and the susceptibilities are given by the following recursion relations:

$$\mathcal{K}^{(l+1)}\{\mathbf{u}\}(m,n) = \frac{\delta_{m,n}}{L} \Big( (\sigma_1^2 C_{R^{(l+1)}}(n) + \gamma) \mathcal{K}^{(l)}\{\mathbf{u}\}(n)$$

$$+ C_{R^{(l+1)}}(n) \sum_{k=2}^\infty \frac{\sigma_k^2}{(n^{(l)})^{k-1}} \sum_{n_{k,1}+\cdots+n_{k,l}=k} \frac{(n_{k,1})!\ldots(n_{k,l})!}{k!} (\mathcal{H}^{(l)})^{*n_{k,1}} \cdots (\mathcal{H}^{(l)})^{*n_{k,l}}(n) \Big). \tag{31}$$

$$\chi_\parallel(m,n)$$
$$= \frac{\delta_{m,n}}{L} \Big( \gamma + C_{R^{(l+1)}}(n)$$
$$\sum_{k=1}^\infty \frac{\sigma_k^2}{(n^{(l)})^{k-1}} \sum_{n_{k,1}+\cdots+n_{k,l}=k} \frac{k(n_{k,1})!\ldots(n_{k,l})!}{(k-1)!} \frac{(\mathcal{H}^{(l)})^{*n_{k,1}} \cdots (\mathcal{H}^{(l)})^{*n_{k,l}}(n)}{\mathcal{K}^{(l)}\{\mathbf{u_0}\}(n)} \Big). \tag{32}$$

$$\tilde{\chi}_\perp(m,n)$$
$$= \delta_{m,n} \Big( \sum_{k=1}^\infty \frac{\sigma_k^2}{(n^{(l)})^{k-1}} \sum_{n_{k,1}+\cdots+n_{k,l}=k-1} \frac{(n_{k,1})!\ldots(n_{k,l})!}{(k-1)!} (\mathcal{H}^{(l)})^{*n_{k,1}} \cdots (\mathcal{H}^{(l)})^{*n_{k,l}}(n) \Big)$$
$$+ \frac{\gamma + 2\tilde{\gamma}\sigma_1}{L^2} \delta_{n,0}\delta_{m,0}. \tag{33}$$

## H SUPPLEMENTARY EXPERIMENTS AND ABLATIONS

Figure 2 compares the statistics of networks with and without a nonlinear activation. In every run we set $C_R = 1$ and examined two width–depth pairs, $\{4, 64\}$ and $\{2, 4\}$. Inputs were sampled from the mean-zero Gaussian random field with covariance $C(x,y) \sim e^{-\|x-y\|^2/K^2}$, and the spectral truncation wavenumber was fixed at 16. For each configuration we initialized 100 independent models. We then plotted, on a logarithmic scale, the kernel value at each frequency, i.e., the squared magnitude of the Fourier transform of the output layer. Each dot represents a single model, while the thick line is the log of the kernel averaged over all 100 models. Upper left figures shows the case with the quadratic activation $\sigma(x) = x^2 + x$. When the activation is absent the kernel drops to zero—within numerical error—exactly after the truncation frequency, but once the activation is applied non-zero energy remains up to twice the truncation frequency, a direct consequence of the second, double convolution term in the recursion formula. Upper right uses $\tanh$, an analytic function, and because every term in its Taylor series contributes, the frequency coupling effect survives at much higher wavenumbers. Lower left employs the non-analytic ReLU; here the kernel decays only slowly beyond the truncation point even if residual connection is absent, which is explained by the theory: the next layer kernel is the Fourier transform of $\rho^{(l)}(x,y)$, and the strong weight of $\rho^{(l)}$ near zero in position space spreads energy across the entire frequency axis. Lower right adds a residual connection with tanh activation; to prevent kernel blow-up the weight variance is scaled by $1 - \gamma$. With a small $\gamma$ the kernel decays rapidly as depth increases, whereas a large $\gamma$ leaves much of the high frequency energy intact, exactly as predicted by the residual–criticality analysis. Across all four figures two patterns are evident. First, increasing the channel width pulls the kernels of individual models ever closer to the ensemble mean, confirming the theoretical $\frac{1}{\text{width}}$ suppression of the four-point vertex. Second, larger widths also reduce the absolute kernel magnitude at frequencies beyond the truncation threshold, again in quantitative agreement with the recursion formula. Now for the comparison between theory and empirical observation, the experimental setup is the same as in Figure 2. We ran separate tests for the tanh, cubic, ReLU and ResNet cases, fixing the channel width at 32 throughout the results are shown in Figure 3. The shaded region in each plot marks one standard deviation across 100 independent runs, and in almost every plot the theoretical curve stays well inside that band. For the tanh and cubic activations we report results at Fourier layer depths 0 and 2; for the ReLU and ResNet cases we show depths 0, 1, 2 and 3. To demonstrate that the agreement is not a coincidence that arises when only a single term is present, we performed ablation tests in which

individual terms were removed from the activation and the theoretical kernel was recalculated. The plots make clear that agreement with the measurements is achieved only when all relevant terms are included in the theory. Lower right figures demonstrate that our theoretical predictions hold not only when $C_R$ is constant with truncation (a step function) but also for more general spectra. To this end we ran an experiment with $C_R(f) = \frac{2.5}{|f|}$ (with truncation), keeping all other settings identical to the constant-$C_R$ case and using the activation function $\sigma(x) = x + \frac{5}{2}x^2 + \frac{50}{6}x^3$. The top panels in lower right figures show results without a residual connection, and the bottom panels with a residual connection. As the plots indicate, the theoretical curves agree closely with the empirical kernels.

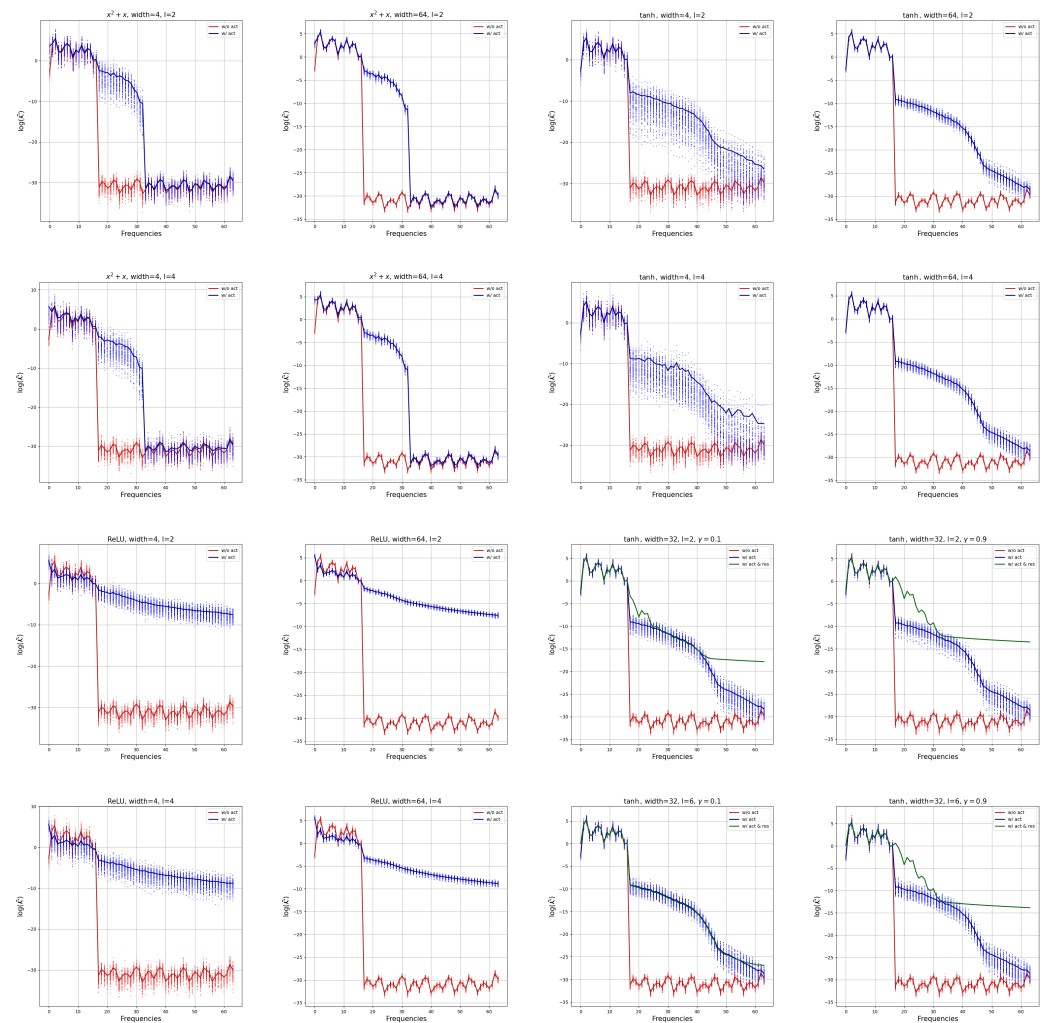

Figure 2: Reduced kernel (log scale) across activations, widths, depths, and residual settings. Top two rows: quadratic (left pair) and tanh (right pair), no residuals, widths $n \in \{4, 64\}$ at depths $L = 2$ (row 1) and $L = 4$ (row 2). Bottom two rows: ReLU (left pair, $n \in \{4, 64\}$) with $L = 2$ (row 3) and $L = 4$ (row 4), and tanh + residual (right pair, $n = 32$) with $\gamma = 0.1, 0.9$ at $L = 2$ (row 3) and $L = 6$ (row 4). All runs use $k_{\max} = 16$ and $C_R = 1$. Dots: individual runs ($N = 100$); solid line: ensemble mean.

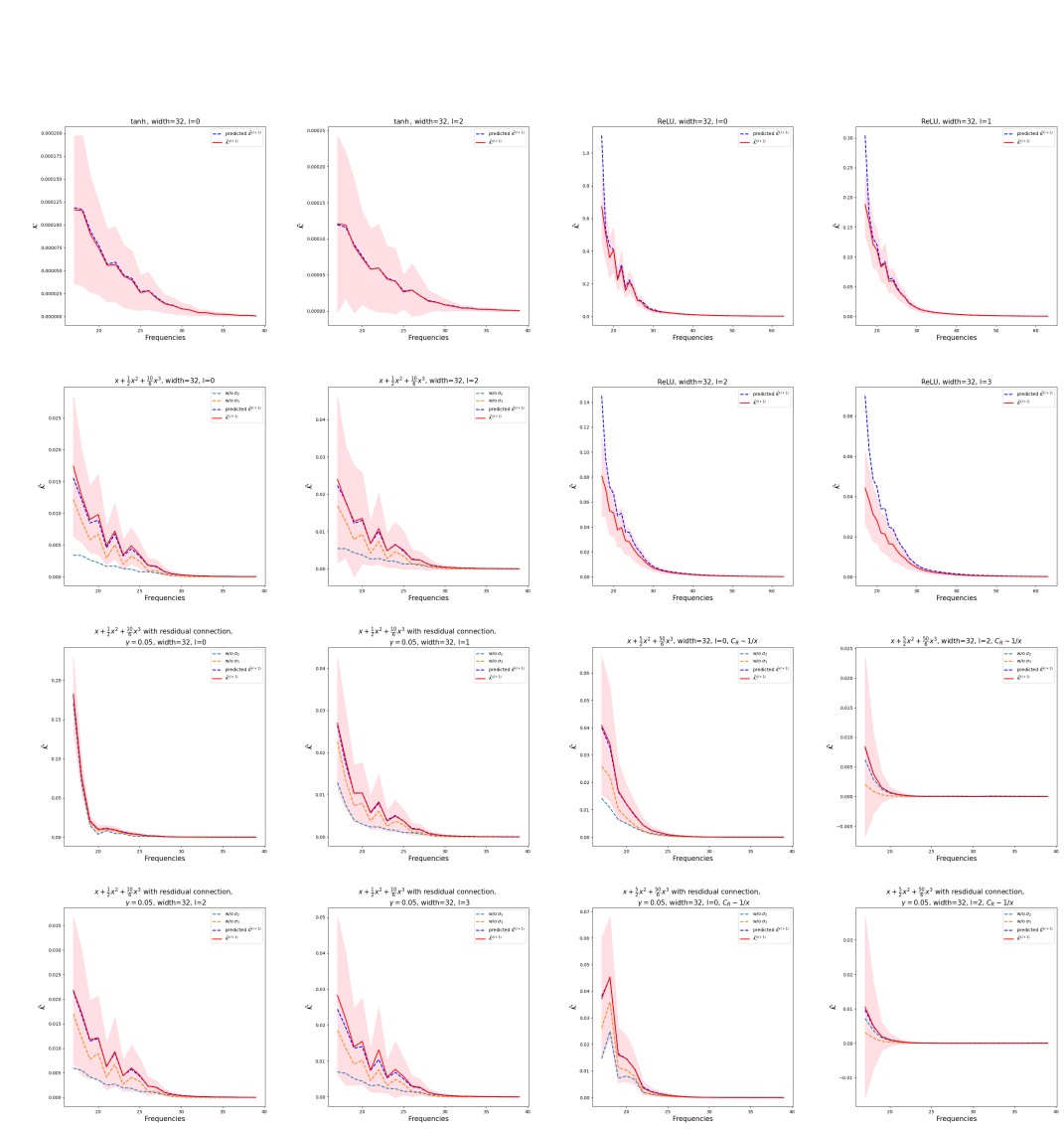

Figure 3: Empirical vs. theoretical reduced kernels (log scale) across activations, depths, and residual settings (width $n = 32$, truncation $k_{\max} = 16$). Row 1: $\tanh$ at $L = 0, 2$; ReLU at $L = 0, 1$. Row 2: cubic at $L = 0, 2$; ReLU at $L = 2, 3$. Row 3: cubic + residual ($\gamma = 0.05$) at $L = 0, 1$; cubic with $C_R(f) \propto 1/|f|$ (truncated at $k_{\max}$) at $L = 0, 2$. Row 4: cubic + residual ($\gamma = 0.05$) at $L = 2, 3$; cubic + residual with $C_R(f) \propto 1/|f|$ at $L = 0, 2$. All non-"decay" panels use a step profile $C_R(f) = \mathbf{1}_{\{|f| \le k_{\max}\}}$. Pink shading: mean $\pm 1$ s.d. over $N = 100$ initializations; solid curve: theory; dots: measurements.

