# OpenReview forum: "Analysis of Fourier Neural Operators via Effective Field Theory"
_ICLR.cc/2026/Conference — ICLR 2026 Conference Withdrawn Submission_

### Official Review · Reviewer_zyvD · 2025-10-28

**Soundness:** 2
**Presentation:** 1
**Contribution:** 1
**Rating:** 2
**Confidence:** 3

**Summary:**

This paper tries to build a theoretical framework for analyzing Fourier Neural Operators (FNOs) using tools from Effective Field Theory (EFT). The authors derive recursive formulas for how the network's "kernel" changes from layer to layer and look at three specific cases. The main claim is that nonlinear activations cause "frequency coupling," creating high-frequency components even from low-frequency input. Experiments are used to check the predictions.

**Strengths:**

1. ​Trying to use EFT to understand FNOs is a theoretically ambitious and interesting goal.
2. The paper shows a serious effort to derive the mathematical formulas carefully, which will be appreciated by theory-oriented readers.

**Weaknesses:**

1. ​**​Unclear Motivation (Major):​**​ The paper's biggest problem is that it doesn't make a strong case for why we need this complex EFT analysis for FNOs. It vaguely says that FNOs lack a "principled explanation" for their behavior, but it doesn't point to a specific, important problem in the existing FNO literature that this theory solves. Why is EFT the right tool for this job? The contribution feels disconnected from practical machine learning challenges.

2. ​**​Hard to Read (Major):​**​ The writing style might be a potential barrier for readers not familiar with EFT. This paper uses very professional words in physics (e.g., "running of couplings," "connected correlators") without properly explaining what they mean in the context of machine learning. The paper might lack of intuitive explanation. What do these recursive formulas mean for someone who wants to build or understand a better FNO? The link between the math and practical outcomes (like better performance or stability) is weak and poorly explained.

3. **Experimental Presentation Needs Improvement (Major):​**​ The clarity of the experimental figures can be enhanced. For example, the verbose titles in Fig. 1's subplots should be condensed and moved to the caption for better readability.

**Questions:**

1. Beyond the general goal of "theoretical understanding," can you give a concrete example of a practical issue in designing or training FNOs that your EFT framework solves, which previous theories could not address?

2. For the broad ICLR audience, could you add a section that explains the core concepts of your theory (like the "kernel recursion" and "susceptibility") in plain language? Specifically, how do these quantities directly relate to performance metrics we care about, like test error?

---

### Official Review · Reviewer_iA2Y · 2025-10-29

**Soundness:** 3
**Presentation:** 3
**Contribution:** 3
**Rating:** 6
**Confidence:** 3

**Summary:**

The paper describes the statistics of infinitely wide Fourier Neural Operators through a complex layerwise recursion involving a "mean metric" whose limit can be described in certain architectures.

**Strengths:**

I am not necesseraly familiar with Effective Field Theory or even Neural Operators, but the authors do a pretty good job explaining the many notions involved.

This paper essentially extends the NNGP analysis for DNNs to Neural Operators, which is novel to my knowledge. The NNGP analysis played a key role in later proving convergence through the NTK and we can hope that this could pave the way to a NTK type convergence result for Neural Operators. It is interesting to see a similar structure emerge as in the NNGP results, where the covariances are described through a recursion involving a complex term that depends on the nonlinearity.

**Weaknesses:**

The formulas are quite big and it can be difficult to get a high level intuition for how these covariances are evolving. Also it would be nice to compare to the NNGP litterature to see if there is any fundamental difference in the Neural Operator settings or whether it is just "the same but in infinite dimension".

The results are restricted to the white noise / uncorrelated Fourier modes case, which kills all correlations between frequencies or spatial correlations (or at least that is what I concluded from equation (7) for example). It feels that some more interesting/unique behavior could emerge if one were to consider a more general setting. Also I might be wrong, but even though it might be true that Neural Operators are initialized with white noise kernels, at the ned of training, one would expect the kernels to become smoother, right? Thus taking smoother kernels at initialization could be closer to the setting we are actually interested in.

**Questions:**

- Are there any fundamental differences between this result and previous NNGP analysis?
- Would the techniques of this paper generalize easily to non-white noise settings?

---

### Official Review · Reviewer_PHQX · 2025-10-31

**Soundness:** 3
**Presentation:** 1
**Contribution:** 2
**Rating:** 2
**Confidence:** 4

**Summary:**

This paper provides a theoretical analysis of Fourier Neural Operators (FNOs) using the framework of Effective Field Theory (EFT). The goal is to explain the stability and frequency behavior of FNOs from first principles. The theory shows that nonlinear activation functions cause "frequency coupling," a process where low-frequency inputs are used to generate new high-frequency information that would otherwise be discarded. For wide networks, the authors derive explicit "criticality conditions" for weight initialization that ensure stable training, providing a guide for hyperparameter selection. These theoretical findings are validated with experiments.

**Strengths:**

1. **Solid Theoretical Analysis**: The paper's theoretical analysis is solid and rigorous. The authors successfully extend the Effective Field Theory (EFT) framework to analyze Fourier Neural Operators (FNOs) in infinite-dimensional function spaces.

2. **Validation of Theory with Experiments**: The authors do not merely present theory but also provide convincing numerical experiments that validate its reliability. The strong consistency between theoretical predictions and empirical results is a key strength of this work.

**Weaknesses:**

1. **Poor Writing and Presentation**: The paper's clarity is a significant issue, to the point of being catastrophic. The dense presentation of mathematical formulas, often without sufficient qualitative explanation, makes it exceedingly difficult for readers, who are unfamiliar with effective field theory, to grasp the core concepts and contributions. The appendix is also problematic, like Appendix H with experimental parameters presented in a long, unstructured format that hinders comprehension. There are also many typos in Appendix. Furthermore, the presentation of the figures could be improved; for instance, the legends in Figures 1-3 are difficult to read due to the small font size.
2. **Limited Originality of Findings**: The central conclusions of this work have largely been established in prior literature. The phenomenon of nonlinear activation functions coupling low-frequency information to high-frequency modes has been well-documented in several papers [1, 2, 3]. The necessity of using a complex tool like EFT to re-derive a known observation is questionable. More critically, the finding of hyperparameter sensitivity (trainability) is nearly identical to [4], which uses the mean-field theory (MFT) framework to derive the same "edge of chaos" criticality condition for stable FNO training. This significant overlap severely undermines the originality of the paper's stability analysis, making it unclear what new insights this work brings to the community.
3. **Limited Contribution Scope and Practical Impact**: Given that the paper's primary findings have already been theoretically demonstrated, the work appears to be "reinventing the wheel" without adding substantial novelty. A more impactful contribution would have been to leverage the this general framework by applying it to analyze and compare different neural operator architectures (e.g., GNO, DeepONet), thereby providing guidance on architectural choices for different PDE problems. Instead, the analysis is confined to FNO. Furthermore, the paper does not adequately bridge the gap between its theoretical findings and practical application. It remains unclear how this framework can be used to effectively guide real-world model design beyond what is already known. The failure to apply the EFT tool to new, un-analyzed problems makes the paper's contributions limited.
4. **Analysis Ignores the Application Context of Solving PDEs**: While the paper presents a theoretical framework, it appears insufficient for explaining the significant performance variance observed when applying different neural operators to various PDE datasets. The analysis does not incorporate properties of the PDE problem itself (e.g., solution smoothness, equation type). By developing a universal theory for the network in isolation, the framework fails to address the crucial interplay between the operator architecture and the specific problem being solved. This disconnect raises questions about whether this is a worthwhile research direction at present, as it does not help practitioners understand why certain architectures excel on certain PDE families and not others.

[1] Convolutional Neural Operators for robust and accurate learning of PDEs

[2] Discretization-invariance? On the Discretization Mismatch Errors in Neural Operators

[3]  Representation Equivalent Neural Operators: a Framework for Alias-free Operator Learning

[4] Understanding the Expressivity and Trainability of Fourier Neural Operator: A Mean-Field Perspective

**Questions:**

See weaknesses.

---

### Official Review · Reviewer_J8jx · 2025-11-01

**Soundness:** 3
**Presentation:** 2
**Contribution:** 2
**Rating:** 2
**Confidence:** 4

**Summary:**

The paper develops an effective field theory (EFT) for Fourier Neural Operators (FNOs), deriving closed-form layerwise recursions for the infinite-width kernel and connected correlators. It shows that nonlinear activations transfer energy to higher frequencies and provides critical-initialization conditions—via parallel/perpendicular susceptibilities—for stable signal propagation, with experiments closely matching the theory.

Contributions:
1) EFT framework for FNOs with explicit kernel/vertex recursions and susceptibility formulas that yield critical (stability) conditions.
2) Explanation and quantification of frequency coupling induced by nonlinear activations, plus analysis of scale-invariant activations and residual connections.
3) Empirical validation: predicted kernel evolution and susceptibilities agree with measurements across activations, widths, and depths.

**Strengths:**

1) Extends effective field theory to FNOs with explicit layerwise kernel/vertex recursions and susceptibility-based criticality conditions.
2) Provides a clear, principled explanation of frequency coupling beyond truncation and analyzes the roles of activation class and residual connections.
3) Technically careful theory with closed-form derivations and internal consistency checks that align with controlled simulations.

**Weaknesses:**

1). Section 4 states the three cases of analytic activations, scale invariant activations, and residual FNOs mostly as closed form recursions without interpretation or usable guidance. The text does not identify which convolution orders dominate low frequency versus high frequency behavior, when kernels grow or decay across depth, or how spectra leak beyond the truncation band. Assumptions on activation coefficients, on the ranges of $\alpha$ and $\beta$, and on residual strength $\gamma$ are not distilled into clear conditions. There are no worked examples that plug a concrete $C_R$ profile to show term by term contributions and crossover regimes. Readers therefore lack regime maps in terms of $C_R$, depth $L$, width $n$, and $\gamma$, as well as rule of thumb designs that translate the theorems into architecture or initialization choices.

2). The experimental evidence is narrow and under specified in the main text. It reports kernel and susceptibility at random initialization with Gaussian field inputs but omits standard trained FNO benchmarks in the main body such as Burgers, Darcy, Poisson, Navier–Stokes, and Helmholtz and provides limited setup details such as grids and resolutions, the rationale for $k_{\max}$, width and depth sweeps, loss and optimizer, and seeds. There are no task level metrics such as $L_2$ error, spectral MSE, or rollout stability, no comparisons to baseline operator learners or standard FNO initializations, and no demonstration that EFT guided choices improve stability or accuracy. Several key hyperparameters such as the $C_R$ profile are relegated to the appendix and code is not linked, which hinders reproducibility.

**Questions:**

1) Section 4 interpretation: after each theorem, what terms dominate at low $k$ and high $k$, and what concrete design rules follow for $C_R(f)$, depth $L$, width $n$, and residual $\gamma$ to avoid kernel growth or collapse through depth.

2) Activation choice: for scale invariant activations with slopes $\alpha$ and $\beta$, provide concrete ranges where frequency coupling is beneficial without causing high $k$ blow up, and compare to an analytic activation such as $\tanh$ on the same task.

3) Classic benchmarks: please add a compact trained section on at least three canonical FNO problems such as 1D Burgers, 2D Darcy, and 2D Poisson or 2D Navier–Stokes or 2D Helmholtz, and report train, validation, and test errors plus spectral MSE to link theory to end to end utility.

4) Baseline comparisons: how does an EFT guided FNO compare to a standard FNO and to another operator learner such as DeepONet or U NO under matched capacity on the same benchmarks.

5) Finite width and depth limits: the theory assumes infinite width. where does it start to deviate at finite width such as $n=16$ or $n=32$, and how deep can you stack layers before higher order effects or truncation errors make the predictions drift.

6) Add a short interpretation right after each theorem. In plain words, highlight which terms dominate at low $k$ and at large $k$, how energy leaks beyond $k_{\max}$, and give simple inequalities on $C_R$, $n$, $L$, and $\gamma$ for stable vs. amplifying regimes. Add one or two line rules of thumb for choosing activation type and residual strength. Include one tiny worked example per case with two spectra — a step profile and $1/|f|$ — show the leading terms at low and high $k$, mark the crossover frequency, and note when truncations are valid.

---

### Note · Authors · 2025-11-12

I have read and agree with the venue's withdrawal policy on behalf of myself and my co-authors.